# Measurement report: The ice-nucleating activity of lichen sampled in a northern European boreal forest

Ulrike Proske[1,2,a], Michael P. Adams[1], Grace C. E. Porter[1], Mark A. Holden[1, 3], Jaana Bäck[4], and Benjamin J. Murray[1]

[1]Institute for Climate and Atmospheric Science, School of Earth and Environment, University of Leeds, Leeds, UK
[2]Institute for Atmospheric and Environmental Sciences, Goethe University Frankfurt, Frankfurt am Main, Germany
[4]Institute for Atmospheric and Earth System Research/Forest Sciences, Faculty of Agriculture and Forestry, University of Helsinki, Helsinki, Finland
[a]now at: Hydrology and Environmental Hydraulics Group, Wageningen University, Wageningen, the Netherlands
[3]Institute for Materials and Investigative Sciences, University of Central Lancashire, Preston, UK

**Correspondence:** Ulrike Proske (ulrike.proske@wur.nl) and Benjamin J. Murray (b.j.murray@leeds.ac.uk)

**Abstract.** Ice nucleating particles (INPs) facilitate heterogeneous freezing of cloud droplets and thus modify cloud properties. Hence, it is important to understand the sources of INPs. During the HyICE-2018 campaign, which took place in the boreal forest of Hyytiälä, substantial concentrations of airborne, heat sensitive biological INPs were observed despite many potential biological sources of INPs being snow covered. A potential source of INPs that were not covered in snow were lichens that grow on trees, hence we investigated these lichens as a potential source of biological INPs in this boreal forest environment. INPs derived from lichen sampled during HyICE-2018 are shown to nucleate ice at temperatures as warm as $-5\,^{\circ}\mathrm{C}$ with $1 \times 10^3$ INPs per gram of lichen. Successive filtration to smaller sizes removes some of the most active INPs in suspension, but substantial activity remains even when filtering to $0.1\,\mu\mathrm{m}$. The small size of the INPs from lichen means they have the potential to either be emitted directly into the atmosphere or be associated with larger particles, such as lichenous reproductive aerosol types (spores, or diaspores). We also show that the INPs from lichens from Hyytiälä are sensitive to heat, which is similar to the INP sampled from the atmosphere of Hyytiälä and consistent with the presence of ice-active proteins. Adding to previous evidence for lichenous INPs, this study shows that lichen from a European Boreal forest in Hyytiälä harbour INPs. This novel finding may be especially important in this snow covered habitat where few, if any, other biological INP sources are available. The great terrestrial abundance of lichens in Hyytiälä, and around the world, calls for further research to combine their ice nucleating ability with dispersal studies to evaluate the flux of lichenous INPs into the atmosphere as well as to what extent these particles reach heights and locations where they might influence cloud properties.

## 1 Introduction

Clouds are a crucial part of the hydrological cycle and strongly affect Earth's radiative balance (Forster et al., 2021). Clouds properties are affected by a range of dynamical and microphysical processes, and it is becoming increasingly apparent that the formation of ice in clouds is amongst one of the least well understood of these processes (Murray et al., 2021; Tan et al., 2016). Heterogeneous freezing of clouds droplets on ice-nucleating particles (INPs) influences precipitation, cloud lifetime and the

radiative effect of tropospheric clouds (DeMott et al., 2010). However, the identity, sources and transport and therefore global distribution of INPs are poorly constrained (Murray et al., 2021). This is especially so for biological INPs, which are known to be active at relatively high temperatures, but are highly variable in concentration (O'Sullivan et al., 2018).

Organic particles from different primary biological origins have been shown to exhibit ice-nucleation ability: biological particles in soil (Conen et al., 2011; O'Sullivan et al., 2014; Hill et al., 2016), on plants (Hill et al., 2014), in sea spray aerosol (DeMott et al., 2016), and in the sea surface microlayer (Irish et al., 2017; Wilson et al., 2015). Primary biological particles including bacteria, pollen, fungal spores, plankton and diatoms have been shown to nucleate ice (Alpert et al., 2011; Schnell and Vali, 1976; Lindow, 1989; Pouleur et al., 1992; Pummer et al., 2012). Subcomponents of living matter have also

been shown to nucleate ice, including cellulose and lignins (Hiranuma et al., 2019; Bogler and Borduas-Dedekind, 2020) . Furthermore, it has been shown that nanometer scale entities washed off fungus and pollen can be potent INPs (O'Sullivan et al., 2016; Pummer et al., 2012; O'Sullivan et al., 2015). In bioaerosols, ice-nucleating ability is a selective property and only few bacterial strains and fungal species have been found to nucleate ice at high temperatures (Hoose and Möhler, 2012; Murray et al., 2012). Globally, more abundant INPs such as desert dust particles dominate the INP distribution at temperatures below

about $-15\,°C$ (Vergara-Temprado et al., 2017). However, biological INPs are thought to have an influence on the hydrological cycle and climate at least on regional scales (Prenni et al., 2009; Spracklen and Heald, 2014; Vergara-Temprado et al., 2017).

While substantial effort has been made to understand the ice nucleating activity of bacteria, pollen and fungus, much less effort has been made to understand the ice nucleating ability of lichens, despite their ubiquity in a variety of environments around the world (Hale, 1974). Several studies have shown that lichens from a range of environments and across multiple

lichen species nucleate ice (Kieft, 1988; Kieft and Ahmadjian, 1989; Ashworth and Kieft, 1992; Moffett et al., 2015; Eufemio et al., 2023). In an early study Kieft (1988) examined 15 lichen. Nearly all of them showed ice-nucleating activity at $-8\,°C$, with $-2.3\,°C$ as the highest onset temperature. The bacteria that could be cultivated from the lichen showed no ice nucleation activity. In a recent study, Eufemio et al. (2023) tested lichens collected across Alaska for their ice nucleating ability, pointing to their possible impact on cloud glaciation in a warming Arctic. Moffett et al. (2015) and Eufemio et al. (2023) between

them surveyed the ice nucleation activity of 86 lichen samples and found that while ice nucleation was ubiquitous these lichens had remarkably varied ice nucleating abilities. Moffett et al. (2015) report onset freezing ranging from $-5.1\,°C$ to $-20\,°C$, while Eufemio et al. (2023) report median freezing temperatures between $-5.2\,°C$ to $-14.5\,°C$. In addition, there is substantial variability in ice nucleation between different samples of the same species of lichen. For example, one sample of Evernia Prunastri nucleated ice at $-5.6\,°C$ while another nucleated ice at $-10\,°C$ (Moffett et al., 2015). These studies show the

ubiquity of ice nucleation in lichens, but given the observed variability in ice nucleating activity, we cannot simply infer that lichens in one environment posses the same ice nucleating activity as the same lichen genus or species in other environments.

For many years lichens were though to be symbiotic organisms composed of a fungal partner, the mycobiont, and a photobiont partner (Nash, 2008). However, it is now recognised that in addition to the mycobiont and photobiont (algae/cyanobacteria), lichen species can accommodate several additional symbionts, including yeasts and bacteria, associated with the fungus

or locally living in the microhabitats of lichen thalli (Aschenbrenner et al., 2016; Cernava et al., 2017; Grimm et al., 2021).

The symbiosis might be seen as a successful one, as lichen are found worldwide, from the tropics to the polar regions (Nash, 2008).

Kieft (1988) concluded that the INPs from the lichen are nonbacterial in origin and suspected them to be either membrane-bound proteins similar to those in bacteria or secondary metabolites. Kieft and Ahmadjian (1989) concluded that lichenous INPs are produced primarily by the mycobiont rather than the photobiont as the former showed ice nucleation activity at warmer temperatures. Kieft and Ruscetti (1990) argued that the sensitivity to proteases, guanidine hydrocholoride and urea could be taken as evidence for a proteinaceous nature of the INPs. In addition, heat treatment has been shown to remove the ice-nucleating activity of lichens (Kieft and Ruscetti, 1990; Daily et al., 2022; Henderson-Begg et al., 2009; Kieft, 1988), which is consistent with the presence of ice-nucleating proteins (Daily et al., 2022; Eufemio et al., 2023). However, ice-nucleating proteins from lichens appear to be more resistant to heat than proteins from bacteria, being stable up to 70 °C (Kieft, 1988). We also know that some fungal materials produce proteins that nucleate ice effectively and these proteins can become separated from the mycelia (O'Sullivan et al., 2015; Schwidetzky et al., 2023a). Ashworth and Kieft (1992) demonstrated ice nucleation activity in whole lichen thalli, whereas in the studies before, the lichen had been ground and brought into suspension. Using a relationship between molecular size and the likelihood to become deactivated on exposure to gamma radiation Kieft and Ruscetti (1992) found a logarithmic relationship between freezing temperature and protein size. This size dependence is consistent with the idea that larger aggregates of proteins have the potential to nucleate ice at higher temperatures (Schwidetzky et al., 2023b).

There are two hypotheses as to why ice-nucleating activity might have evolved in lichens (Kieft, 1988): Firstly, as proposed for ice nucleation active bacteria, lichen might benefit from nucleation of ice at relatively modest supercooling and the more gradual formation of ice as it is less stressful to an organism than rapid crystallisation experience in greater supercoolings. During rapid freezing at great supercooling intracelular ice formation becomes more likely and this is usually lethal to cells (Clarke et al., 2013; Daily et al., 2020, 2023). Secondly, ice nucleation might be a water-harvesting mechanism (Kieft, 1988; Henderson-Begg et al., 2009). Once a small amount of water is frozen on the thallus, more water may preferentially deposit on it. Later, when the temperature increases, this ice may melt and the liquid water would become available to the lichen. This process is all the more important, since lichen lack stomata and are therefore not able to actively control water loss as many plants do (Kappen and Valladares, 2007).

The distribution of aerosol particles originating from lichen in the atmosphere is poorly constrained. Lichens are complex and varied organisms that have evolved to produce entities that can become airborne. To appreciate which components of lichen might become airborne (and therefore design an appropriate ice-nucleation study) we need some understanding of the forms and structures of lichens. Three growth forms of lichens are traditionally distinguished (Hale, 1974): Crustose lichen are in intimate contact with their substrate and cannot readily be separated from that substrate. Foliose lichen have a leafy plant body, up to 0.3 m in diameter. Fruticose lichens appear hair-like. Lichen reproduce either sexually or asexually, producing spermatia (1 to 5 μm), spores (1 to 510 μm) or vegetative diaspores (10 to 3000 μm, e.g. isidia or soredia (Hale, 1974)). Soredia are powdery granules of algae cells enveloped by fungal threads, wheras isida are spiny outgrowths that are easily broken off the thallus (Hale, 1974). While spores are forcibly ejected, vegetative diaspores rely on external forces to be removed from the

thallus (Bowler and Rundel, 1975). The vegetative strategy allows the invasion of new habitats, and species using this strategy often have a greater world distribution than their sexual counterparts (Hale, 1974; Bowler and Rundel, 1975). Clearly, these particles also have the potential to contribute to the INP population of the atmosphere if they harbour ice nucleating entities.

As of now, little is known about the atmospheric abundance of ice nucleation active particles that stem from lichen (Després et al., 2012). However, the wide distribution of lichen indicates that they have a successful mechanism for dispersal (Hale, 1974), which in turn suggests high concentrations of propagules (e.g. spores, soredia and isidia). Wind removal of soredia has been successfully demonstrated by Bailey (1966). It was found that while more soredia were removed at higher moisture content of the thallus, the wind speed necessary for removal increased with higher moisture content as well. Armstrong (1993) conducted a wind tunnel experiment and found humidity to lead to a substantial decrease in soredia dispersal. Armstrong (1991) identified humidity to be the most significant variable for soredia dispersal, stating that more soredia dispersed at lower humidity. It has been suggested that high moisture content promotes soredia production, but that low moisture content facilitates release (Marshall, 1996). In a year-long aerobiological monitoring program over Antarctica, Marshall (1996) found lichen soredia to be more abundant than spores. Soredia were collected every month, suggesting that soredia are produced year-round. The release was found to be independent of any specific meteorological variable. In a field study in an old-growth forest in Finland, samples of the lichen Lobaria pulmonaria were genetically analysed (Ronnås et al., 2017). Symbiotic propagules had a maximum dispersal range of $100\,\mathrm{m}$, while ascospores dispersed several kilometers. Tormo et al. (2001) placed traps $6\,\mathrm{m}$ above ground level in Spain. The mean soredia concentration was $0.4\,\mathrm{m}^{-3}$ with a daily maximum of $5\,\mathrm{m}^{-3}$. The concentration was higher during the day. Positive correlations with temperature and again negative correlations with humidity were found.

In this paper we report the ice nucleating ability of lichens that were present and exposed in the boreal forest at the Station for Measuring Ecosystem-Atmosphere Relations (SMEAR) II located in Hyytiälä, Finland during the HyICE-2018 campaign. The HyICE-2018 campaign was focused on measuring atmospheric INPs between February and June 2018 (Brasseur et al., 2022). Schneider et al. (2021) report the presence of heat sensitive biological INPs between $-25$ and $-6\,°\mathrm{C}$ during the campaign. In addition, Vogel et al. (2024) suggested that INP active below $-24\,°\mathrm{C}$ are related to a biological source. However, the source of these INPs was unclear since the surface was snow covered, which rules out leaf litter or bare soil as sources of INPs. We hypothesize that the trees which bore lichen, that were exposed even in the winter when the canopy and the ground were snow covered (see Fig. 1), might have been a source of INPs. In this paper we tackle the first part of the hypothesis, namely the question of if the lichens in the forest during the winter of HyICE-2018 contained ice-nucleating entities. Eufemio et al. (2023) have made this possibility obvious since they showed that lichen from across Alaska harbour INPs. It remains to us to confirm this for the Hyytiälä boreal forest, and to investigate the size of the lichenous INPs that we find. A positive outcome would provide motivation to address the question of whether sufficient quantities of INPs are released into the atmosphere to influence the INP population and subsequent cloud formation.

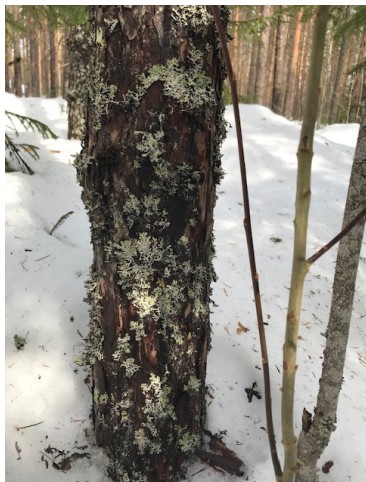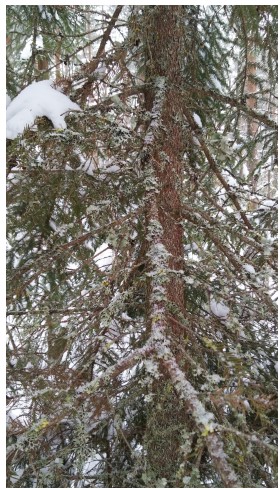

**Figure 1.** Lichens attached to trees are ubiquitous in Hyytiälä and remain exposed even when the ground is covered in snow (first photograph taken on 07.04.2018 and second on 22.03.2018).

## 2 Methods

In order to quantify the ice nucleating ability of lichens that are potentially relevant as sources of atmospheric INPs we collected samples and analysed their ice nucleating activity in the immersion mode using a droplet freezing assay. Samples were collected

and as discussed below were sampled as mixtures of multiple lichen species. Our experimental approach was to subsample from these mixtures of lichen and test the activity of the mixtures of species. We examined mixed samples of lichen for their ice nucleating ability, size of the ice-nucleating species and the heat sensitivity rather than solely focusing on single species in order to reveal if the lichens in Hyytiäla harboured ice nucleating entities and obtain an indication of their activity. This allowed us to address our stated objective of determining if there is a potential source of biological INPs associated with the

prolific lichen population in Hyytiäla. In addition we performed a set of experiments where we attempted to separate the lichen species.

### 2.1 Sample collection and identification

The lichen was taken from Scots pine trees in a boreal forest environment in Hyytiälä with clean tweezers and placed in resealable plastic bags in March and April 2018 during the HyICE-2018 campaign. Lichen samples were imaged using a

stereomicroscope (Stemi 508, Zeiss) and were identified with these pictures (see Fig. 2 and 3). One plastic bag contained specimen of *Evernia prunastri* (foliose), *Bryoria sp.* (fruticose) and *Platismatia glauca* (foliase), and two other plastic bags were filled with *Hypogymnia physodes* (foliose). The species were not collected separately because they tended to grow in the same locations and clear separation in the field by eye was challenging (or impossible). These bags were sealed and transported back to the University of Leeds on a passenger flight, at room temperature. Experiments were conducted in April and May 2018.

By storing at room temperature, the samples were preserved at low relative humidity, conditions under which biological activity

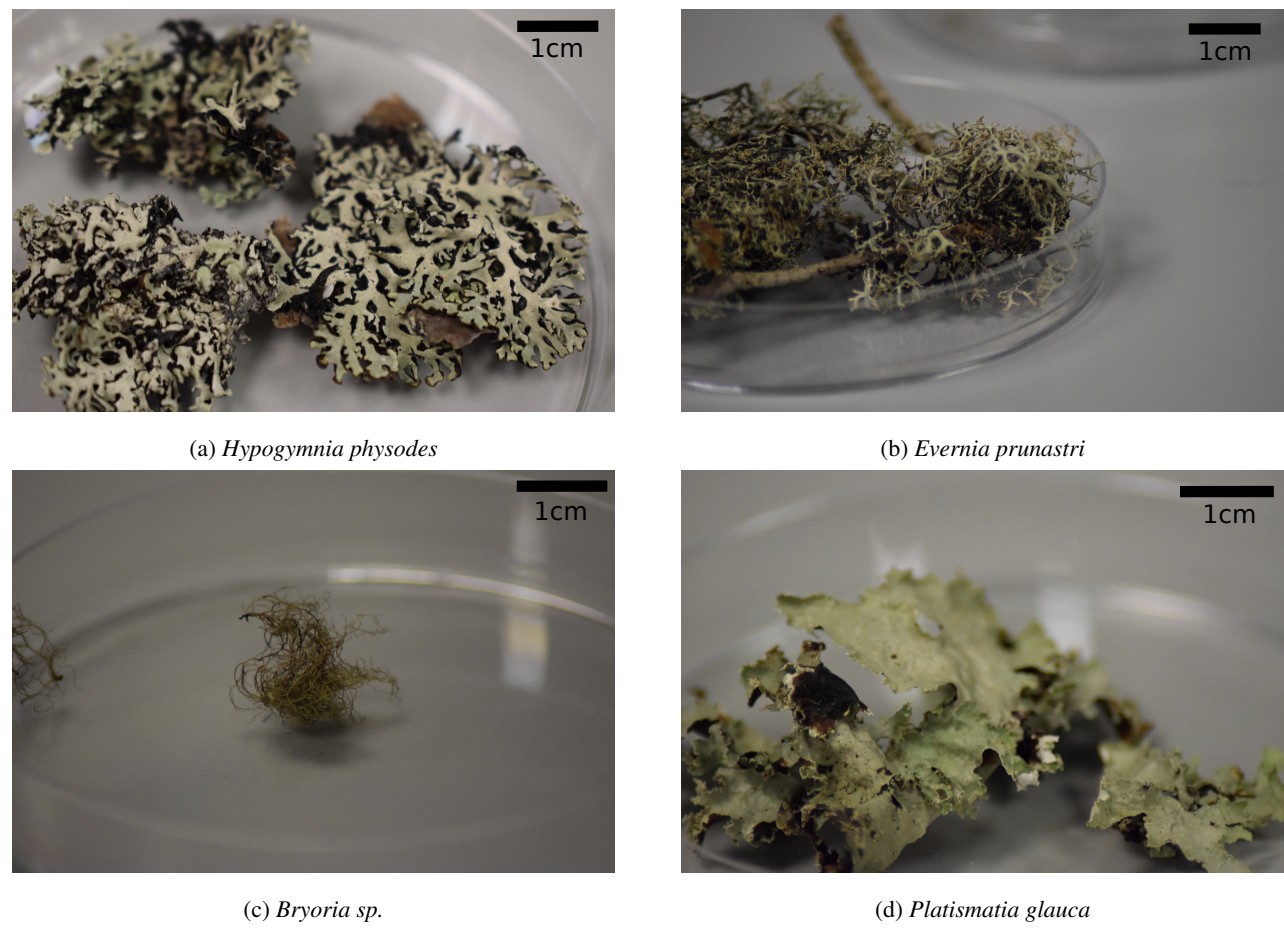

(a) *Hypogymnia physodes*

(b) *Evernia prunastri*

(c) *Bryoria sp.*

(d) *Platismatia glauca*

**Figure 2.** Pictures of the sampled lichen species that were tested for ice nucleation activity in this study.

is inhibited. Nevertheless, we note that the storage at room temperature was pragmatic, and it is possible that the activity of the samples might be somewhat dependent upon the storage conditions.

Examples of the structures that can become aerosolised are shown in Fig. 3. Panel a shows soredia on *Platismatia glauca*, while panel b shows isidia on *Evernia prunastri*. These vegetative diaspores can be broken off the thallus through the action
of wind, rain droplets or even animals. The recognition that it is likely these fragile structures on the surface of the lichen that preferentially become airborne helped us to design a droplet freezing assay that is appropriate. In some previous ice nucleation studies the lichens were ground with some water to produce a pulp that was then suspended in water (Kieft, 1988; Eufemio et al., 2023). This approach might be appropriate for studying the water harvesting properties of lichens, but may be less relevant for understanding atmospheric implications. In addition, the practice of washing lichen samples to remove non-lichen
ice-nucleating entities may inadvertently remove the soredia and isidia, the very entities of particular interest. Hence, we used an approach where lichen was exposed to water and gently agitated in order that fragile structures, like the soredia or isidia, might be removed (details in the next section). The large pieces of lichen were allowed to settle to the bottom of the vial and

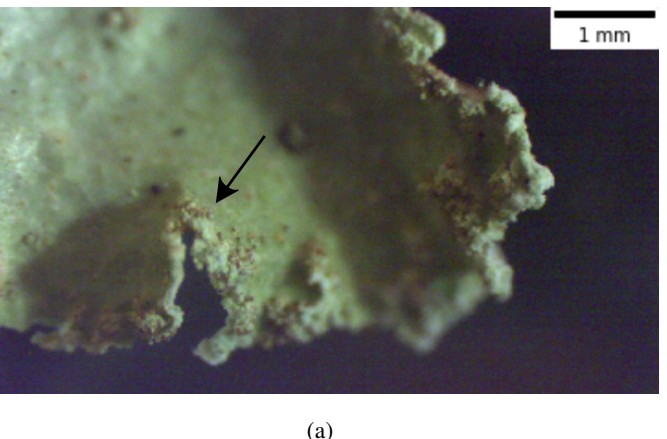
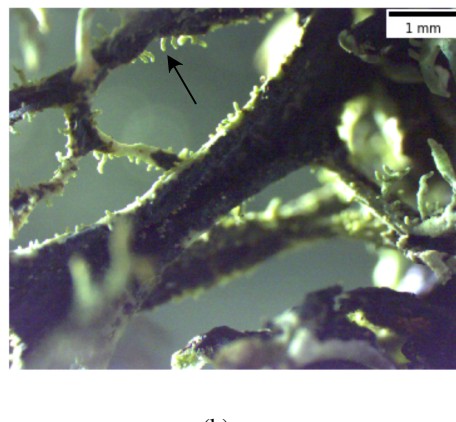

(a) (b)

**Figure 3.** Images obtained by stereomicroscopy of the sampled lichen that were immersed in water for the species specific testing. **a)** Soralia (see arrow) of *Platismatia glauca* that produce soredia, **b)** isidia (see arrow) of *Evernia prunastri*.

then the aqueous supernatant, which was clear to the eye, was sampled for the droplet freezing assay. While this approach is clearly different from bioaerosol production from lichen via wind, it does bias the analysis towards the entities associated with
lichen that are likely to become aerosolised.

### 2.2  Sample preparation

In order to generate clean water with minimal ice nucleating contamination, $50\,\mathrm{ml}$ of nanopure water was filtered through a $0.2\,\mathrm{\mu m}$ filter (Minisart®, sartorius) and deposited into a $50\,\mathrm{ml}$ polypropylene test tube. From this water, a blank was run on the Microlitre Nucleation by Immersed Particle Instrument (µl-NIPI, see section 2.5, Whale et al. (2015)) each morning to establish
the baseline in the lab. In this way, the background signal could be evaluated, which determines the limit of detection for the instrument. The INPs examined in this study had freezing temperatures in between $-5$ and $-28\,^{\circ}\mathrm{C}$ (except for few dilutions where the freezing temperature was somewhat lower), with $T_{50}$ approximately in between $-15$ and $-20\,^{\circ}\mathrm{C}$. Therefore, a blank was regarded as acceptable if the first droplet froze at below $-22\,^{\circ}\mathrm{C}$, $T_{50}$ was at below $-26\,^{\circ}\mathrm{C}$ and the last droplet froze at below $-30\,^{\circ}\mathrm{C}$. The water from which the blank had been taken was then used further for immersing the lichen or for dilutions
of suspensions. Samples that froze below $-25\,^{\circ}\mathrm{C}$ and thereby went into the range of considerable blank freezing (see Fig. A1), were excluded from the analysis.

To prepare the aqueous lichen extracts, lichen was taken from its plastic bag with tweezers, separated from any bark or twigs, and placed in $50\,\mathrm{ml}$ polypropylene centrifuge tubes (Falcon tubes, Fisher Scientific). The lichen's mass was noted as well as the amount of water which was then added to the tube. Lichen samples were taken from a mixture of lichen species stored in
a single sealed plastic bag. Both sample B and C had a similar proportion of the different lichen species, but the sensitivity to the mixing method was explored (see Sec. B and Fig. B1 for results). The composition of the mixtures and the dates on which the samples were immersed in water and run on the µl-NIPI are given in table 1. The composition was meant to mimic

| Sample name | Species present | Concentration ($\mathrm{g\,ml^{-1}}$) | Extraction method | Dates of µl-NIPI runs |
|---|---|---|---|---|
| B | *Evernia prunastri* ($\sim 40\%$), *Bryoria sp.* ($\sim 10\%$), *Platismatia glauca* ($\sim 40\%$) | 0.030 | 10 mins by hand | 24.04., 25.04.18 |
| C | *Evernia prunastri* ($\sim 40\%$), *Bryoria sp.* ($\sim 10\%$), *Platismatia glauca* ($\sim 40\%$) | 0.033 | 30 mins on rotary mixer | 02.05., 03.05., 04.05.18 |
| Species specific tests | Every species separately | | 30 mins on rotary mixer | 09.05., 10.05.18 |

**Table 1.** Overview of the lichen samples that were tested during this study.

the mixture in the bag. For the species specific runs, samples of only one species of lichen were put into each centrifuge tube. Sample B was rotated carefully by hand for $10\,\mathrm{min}$, while sample C and the species specific samples were left on a rotation mixer for $30\,\mathrm{min}$, set at about 30 rotations per minute. In all procedures, care was taken to use relatively gentle approaches so as to minimise the break up of any structures in the lichen, since the atmospherically relevant INPs should be on its surface and readily removable. Sample B is split into B1 and B2 in the manuscript, B2 was sampled from the same suspension one day later, so had had more time to release INP into suspension/solution.

## 2.3 Filtration

In order to learn more about the size of the ice-nucleating entities, samples were also filtered prior to testing. $10\,\mathrm{\mu m}$ (NY10, Merck Millipore), $2\,\mathrm{\mu m}$ (TTP, Isopore™, Merck Millipore), $0.1\,\mathrm{\mu m}$ (6809-6002, Anodisc, Whatman®) and $0.02\,\mathrm{\mu m}$ (6809-6012, Anodisc, Whatman®) filters were used, placed within a $45\,\mathrm{mm}$ Advantec 301000 stainless steel filter holder. The filters were employed in front of a syringe filled with sample. Only for sample C all filter sizes were used, as it was realized after the processing of sample B that further size differentiation would be desirable. For the species specific tests the samples were partly too small to use all filters so only the $2\,\mathrm{\mu m}$ filter was used. It should be noted that the size of the particles immersed in water may be different from the size of the dry particles that might become aerosolised.

## 2.4 Heat Test

To test whether the ice nucleating particles are heat-labile, a polypropylene test tube with $1\,\mathrm{ml}$ of sample solution was placed in a boiling water bath for $30\,\mathrm{min}$. Different temperature heat treatments have been shown to have a different effect on biological INPs, and more deactivation happens with higher temperatures (Hara et al., 2016). In most studies, heat treatments that are meant to test for biological INPs involve heating at about $90\,^\circ\mathrm{C}$ for $10\,\mathrm{min}$ (Hara et al., 2016; Christner et al., 2008; Moffett

et al., 2015; O'Sullivan et al., 2014). Longer periods of 20 and 45 min have been used as well (Garcia et al., 2012; O'Sullivan et al., 2015). In this study, the sample containers were placed in a bath of boiling water, hence the sample was warmed to above $90\,°C$, for 30 min as recommended by Daily et al. (2022). The sample was left to cool for a few minutes before being run on the µl-NIPI. Daily et al. (2022) have shown that not only biological INPs but also some minerals are affected by the wet heat tests, but since our samples are clearly biological in nature we regard it as a valid method for the qualitative detection of protein based biogenic INPs.

## 2.5  Ice nucleation measurements with the Microlitre Nucleation by Immersed Particle Instrument

Droplet freezing techniques are widely used to study immersion mode freezing (Vali, 1971; Murray et al., 2012). In these techniques, droplets of a suspension are cooled and freezing events are recorded as a function of temperature. The volume of the droplets determines the temperature range that can be investigated. In a bigger volume of the same concentration there are more INPs and therefore it is more likely that rarer INPs are present in the droplet, which are active at higher temperatures and dominate the freezing. Hence, multiple instruments with different droplets sizes and dilutions are needed to investigate ice-nucleating particles and their range of freezing temperatures.

The Microlitre Nucleation by Immersed Particle Instrument (µl-NIPI) is used for droplet freezing experiments with $1\,µl$-droplets. As such it was first described by Atkinson et al. (2013) and has since been employed in a range of studies (O'Sullivan et al., 2014, 2015). It has been part of an intercomparison between 17 ice nucleation measurement techniques in Hiranuma et al. (2015) and DeMott et al. (2018). In Whale et al. (2015), one can find a detailed instrument description.

For the freezing experiments, a $22\,mm$ diameter silanised glass slide (Hampton Research, HR3-231) is put on the aluminium plate after being rinsed thoroughly with methanol and nanopure water. If the samples had been prepared hours before being tested, they were vortexed before being run on the µl-NIPI in order to homogenize the sample and stir up any particles that might have sedimented. Samples that were run in Leeds were vortexed for at least $10\,s$ before each run. 30 to 50 $1\,µl$-droplets were then pipetted directly onto the slide using a multi dispense pipette (Sartorius eLINE®). Suboptimal mixing before pipetting is visible in the results where e.g. dilutions do not line up, as discussed in Sec. 3. The aluminium plate was covered with a Perspex chamber. The chamber has openings for a camera (Microsoft LifeCam HD) and two pipes for flushing with dry nitrogen. The nitrogen flow prevents frost growth and freezing due to contact with frost. Another benefit is that the nitrogen flow reduces potential contamination with aerosol particles from laboratory air.

The cold stage (EF600 Stirling engine chiller, Grant-Asymptote) was directed to cool with a temperature ramp of $1\,°C\,min^{-1}$. The starting temperature was between $10\,°C$ and room temperature and the run was stopped as soon as the last droplet was observed to be frozen. When the EF600 was set to cool, the data logger software was started. The temperature of the aluminium plate as a function of time was logged, and images of the droplets on the glass slide were recorded with the digital camera at a rate of $1\,frame\,s^{-1}$. The frames of the video were manually looked through, noting for each droplet the frame in which it showed the first signs of freezing so the fraction of droplets frozen, f, could be calculated. The ice active site density per mass

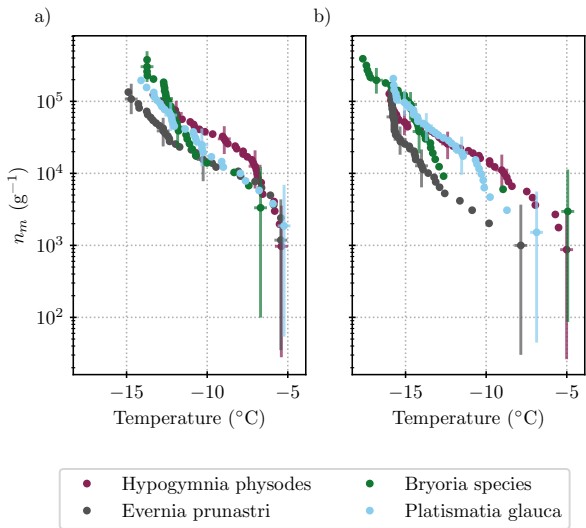

**Figure 4.** INP measurements that were conducted for each species of lichen separately: comparison of the a) unfiltered samples and of b) those that were filtered through a 2 μm filter. For a comparison with literature values see Fig. 7.

of lichen, $n_m$, was calculated following Murray et al. (2012) (who reference Vali (1971)):

$$n_m = -\frac{\ln(1-\mathrm{f})}{V_d} \cdot d \cdot \frac{V_w}{m_{\mathrm{lichen}}} \tag{1}$$

where the factor $d$ is the dilution of the droplets, $V_d$ is the droplet volume, $V_w$ is the wash volume, and $m_{\mathrm{lichen}}$ is the mass of lichen per sample. We normalise to the initial mass of lichen in order that we can quantify the relative changes in activity on dilution, heat tests and filtration. Dilutions of samples were made in order to reach lower temperature ranges in the measurements. The errors of the INP concentration measurements were calculated following the procedure described in Harrison et al. (2016), which in turn is based on Wright and Petters (2013), and the temperature uncertainty is $\pm 0.4\,°\mathrm{C}$.

## 3 Results

Previous studies have shown that a substantial fraction of the INP observed during the HyICE-2018 campaign was of biological origin, based on a heat test (Schneider et al., 2021). Our hypothesis is that these biological INPs originate from the lichen that is abundant in the boreal forest ecosystem even when there is snow cover. However, because INP typically make up a small portion of aerosol particles, their identification in aerosol samples is challenging. Thus, samples of lichen were taken in Hyytiälä and tested for their ice nucleation activity.

| Species | Mass of lichen (g) | Washvolume (ml) | Concentration (g ml$^{-1}$) |
|---|---|---|---|
| *Hypogymnia physodes* | 0.8709 | 30 | 0.029 |
| *Evernia prunastri* | 0.9109 | 35 | 0.026 |
| *Bryoria sp.* | 0.0185 | 2 | 0.0093 |
| *Platismatia glauca* | 0.3515 | 20 | 0.018 |

**Table 2.** Lichen concentrations in the species specific tests

## 3.1 Ice nucleating ability of the individual lichen species

Fig. 4 shows the ice nucleation activity of each lichen species that was tested individually. The species specific ice nucleation activities look very similar to each other. Only the spectrum of *Hypogymnia physodes* has a shape distinct from the others, with a greater activity below $-7\,°C$ and a lower activity above $-7\,°C$. One might conclude that the three species that were stored in one bag (*Evernia prunastri*, *Bryoria sp.* and *Platismatia glauca*) and were in close proximity to one another in the forest simply show the same INP concentrations because entities such as the soredia and isidia may have been spread throughout the sample. However, the $2\,\mu m$-filtered size fractions, shown in Fig. 4 is inconsistent with this idea. *Evernia prunastri* and *Platismatia glauca* are the species that were present the most in the bag as well as in the mixed samples (see table 2). The INP concentration of the *Platismatia glauca* sample in the $2\,\mu m$ size fraction is half an order of magnitude higher than that of the *Evernia prunastri* sample at for example $-10\,°C$, while the spectra of the unfiltered samples lie within error of each other. So in fact, the different lichen species seem to harbour differently sized INPs.

## 3.2 Mixed lichen samples

Further filtration and heat tests were done on mixed samples of the lichen species. Figure 5 shows all measurements that were made on lichen sample B, a mixture of *Evernia prunastri* and *Plasmatia glauca* with only a small fraction of *Bryoria sp.* (see table 1).

In Figure 5a), the ice nucleation measurements that were conducted on different size fractions of sample B1, are plotted together for comparison. The spectra for the unfiltered sample and the sample that had been filtered through a $10\,\mu m$ filter look similar, qualitatively and quantitatively (within $2\,°C$). This indicates that the ice-nucleating entities can become independent of the lichen and that these entities are mostly smaller than $10\,\mu m$. The ice-nucleation activity drops by 1 to 2 orders of magnitude (a shit in temperature of about 4 to $5\,°C$) at respective temperatures for the sample that had been filtered through a $0.1\,\mu m$ filter. This suggests that the size of most of the entities that nucleate ice above -18°C falls in between $0.1\,\mu m$ and $10\,\mu m$. In order to better constrain this size, a $2\,\mu m$ filter sample was included in the next set of experiments (sample C, see below).

A striking feature that is present in all three size fractions of sample B1 are the 'steps' in the spectra (sections that are almost vertical), visible at $-16\,°C$ and $-18\,°C$ (see also the differential spectra in Fig. C1). A sharp rise in INP concentrations at a specific temperature suggests that there is a single INP species with a specific temperature of freezing onset present in the

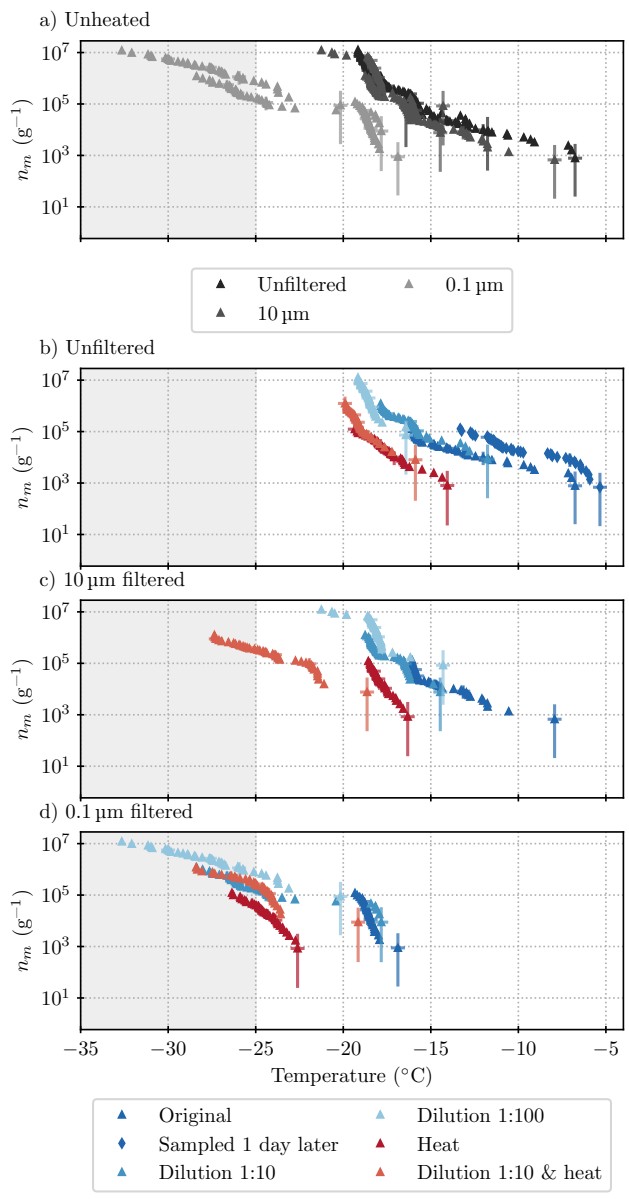

**Figure 5.** Results acquired from all ice nucleation measurements on lichen sample B (INP per gram of lichen sample): **a)** all unheated samples (summarizes all blue points from the plots below), **b)** unfiltered samples, **c)** samples filtered through a $10\,\mu m$ filter, **d)** samples filtered through a $0.1\,\mu m$ filter. Higher dilutions are depicted as lighter shades of blue, heat tested samples in red. The grey shading indicates the temperature at which the blanks started to freeze and thus results are deemed unreliable. The majority of results are for sample B1, but there is one run with B2 in panel b. Uncertainties are included for every 10th data point and deduced as described in Sec. 2.5.

sample. As there is no data for the concentration of the INP species active at or above −16 °C for the 0.1 μm size fraction, it can be concluded that this entity is between 0.1 μm and 10 μm in size. The species active at −18 °C, however, is also detected in the 0.1 μm size fraction.

All heat tests conducted on these samples show a striking loss of activity, with a decrease in the temperatures where ice nucleation was observed. The onset freezing temperature that could be observed with 1 μl droplets dropped by 6 °C to 8 °C, depending on the size fraction. In fact, the heat test decreased the concentration of INPs active at temperatures greater than -14 °C so much that they could not be detected with the μl-NIPI technique. The loss of activity is greatest for INPs active at temperatures above about−16 °C; being more than an order of magnitude. INPs that induce freezing at lower temperatures

seem to be less heat-labile. In the unfiltered and 10 μm size fraction, after the heat test, activity is centred around −18 °C, just below the step at −18 °C (see Fig. 5 b) and 5c)). However, in the 0.1 μm size fraction (see Fig. 5d)), the heat test diminishes activity at −18 °C. Therefore, the INPs that are active around −18 °C and dominate the heated unfiltered and 10 μm samples are concluded to be larger than 0.1 μm. The spectrum of the heat test in the smallest size fraction, 0.1 μm, is very close to the unheated sample spectrum below −24 °C, indicating that the INP that are smaller than 0.1 μm and active below −24 °C are

heat stable (although these results are very close to our background). Overall, the heat tests show that the INPs that are active at temperatures warmer than −18 °C, including those species responsible for the steps at −16 °C in the unheated spectra, are heat-labile. In the heated samples larger than 0.1 μm, INPs that are active at around −18 °C, dominate. It seems likely that the species active at −18 °C is only partially heat-labile or in fact consists of two species: it dominates in the heated samples of size fractions larger than 0.1 μm, but its activity diminishes upon heating of the 0.1 μm size fraction.

In order to test if the activity of sample B changed with time, we took a sample of aqueous solution (B2) from the original centrifuge tube that contained water and lichen. This sample had been rotated by hand prior to B1 being sampled, and then left for 1 day in a fridge (at 4 °C) before sample B2 was taken. The activity in this sample was up to about an order of magnitude greater than in sample B1 (see Fig. 5b)). A potential explanation for this is that the lichen shed additional INP into solution while the lichen was sat in water. However, we note that precipitation samples have been seen to become more active with

time, possibly related to the formation of ice-active protein aggregates (Stopelli et al., 2014). While this is in itself interesting and warrants future work, it also informed us that we could not perform more freezing assays with the original lichen-water mixture for further tests (such as additional filter tests). Hence, it was necessary to make up fresh suspensions.

In order to investigate the size of these INPs present in the lichen samples further, a new sample of lichen (sample C) was taken out of the same bag as sample B. When sampling from the bag of lichen, we aimed to obtain the same mix of lichen

species in this sample as in sample B (see Table 1). Figure 6 shows the results of the experiments conducted with sample C. Inspection of Figure 6a shows that while the freezing characteristics are qualitatively similar to sample B there are also some important differences that we discuss here.

Again warm temperature INPs are present, and as in sample B, INP concentrations decreases substantially when the sample is filtered through a 0.1 μm filter (Figure 6a) and Fig. C1). Meanwhile, the 2 μm size fraction gives about the same signal as the

295 10 μm fraction. The bulk of the INPs present in the mixture of lichen species in lichen sample B and C are therefore concluded to be in between 0.1 μm and 2 μm in size. The steps that can be clearly distinguished in all three size fractions of sample B1

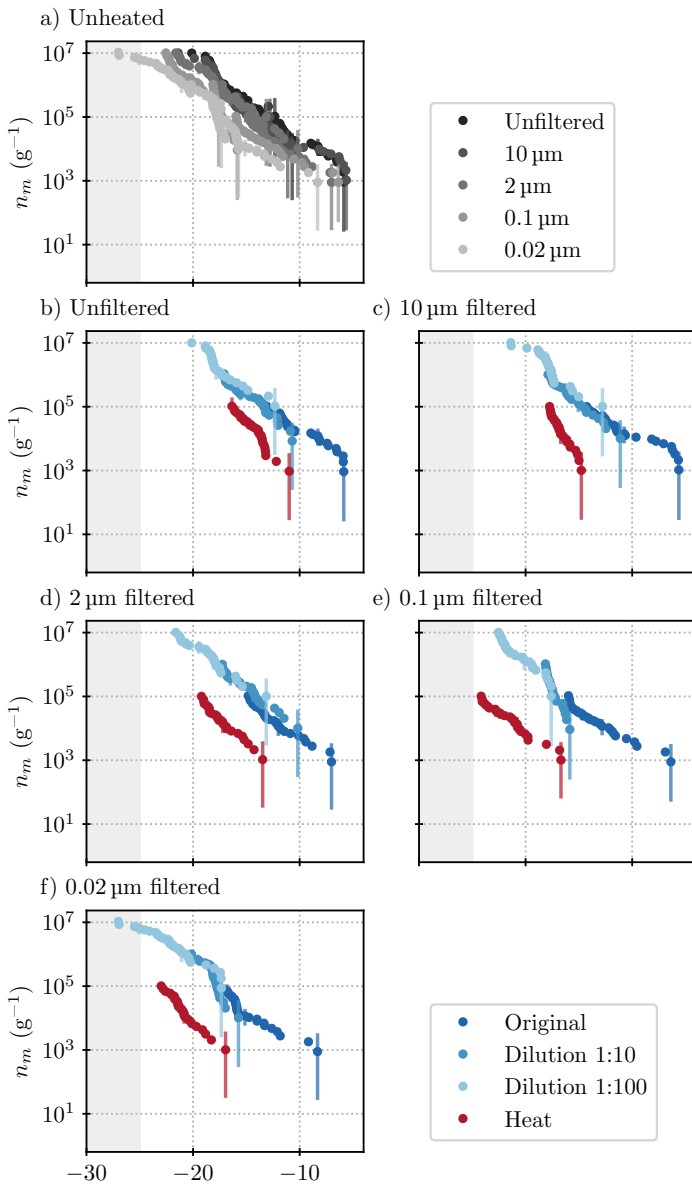

**Figure 6.** Results acquired from all ice nucleation measurements on lichen sample C: **a)** all unheated samples, **b)** unfiltered samples, and samples filtered through a **c)** 10 µm, **d)** 2 µm, **e)** 0.1 µm, and **f)** 0.02 µm filter. Higher dilutions are depicted as lighter shades of blue, heat tested samples in red. The grey shading indicates the temperature at which the blanks started to freeze and thus results are deemed unreliable.

are less prominent in sample C (but visible in the differential spectra in Fig. C1). The step at $-18\,^\circ$C can be seen in all size fractions, albeit naturally at lower concentrations in the $0.1\,\mu$m and $0.02\,\mu$m size fraction (because higher temperature INP are removed by filtration and this reduces the concentration across the full spectrum since this is a cumulative quantity). The step at $-16\,^\circ$C, however, can only be identified in the $0.1\,\mu$m and $0.02\,\mu$m size fraction (see Fig. 6e) and 6f)). Only when the larger INPs are removed can the activity of these INPs be seen in the spectrum, because their concentrations are lower than that of the INPs larger than $0.1\,\mu$m. That the step at $-16\,^\circ$C was visible in the spectra of sample B1 might be explained by the differing composition of the samples.

The heat tests on sample C again show a clear decrease in activity, by about 1 to 2 orders of magnitude (a shift of about 5 to $10\,^\circ$C. The higher temperature INPs are more heat-labile than the ones active at colder temperatures, just like in sample B1. Also, as before, in the $10\,\mu$m size fraction in Fig. 6c), an INP species active at $-18\,^\circ$C is evident in the spectrum of the heated sample. However, heat tests of the smaller size fractions diminish activity at $-18\,^\circ$C.

Taking the results from lichen sample B1 and C together (where the extraction methods were different; see methods and Sec. B and Fig. B1 for results), there is an INP species active at $-18\,^\circ$C present in all samples, whose concentration decreases when filtered through $2\,\mu$m pore size filters or smaller, but remains distinguishable in the spectra of the $0.02\,\mu$m size fraction. The species active at $-18\,^\circ$C is partially heat-labile as a step at $-18\,^\circ$C is also visible in the heated samples, but only in the size fractions larger than $2\,\mu$m. These different characteristics (activity and heat sensitivity) for different size fractions point to different states of the INP species, for example as either attached to a larger particle or free in solution as it has been proposed for nano-INP (O'Sullivan et al., 2015) or as different aggregates as proposed for Pseudomonas syringae (Turner et al., 1990). The INP species active at around $-16\,^\circ$C is heat-labile and smaller than $0.02\,\mu$m for sample C (see Fig. 6). The identification of two INP species, active at certain temperatures and of a determined size, allow the relation of these findings to future studies that might find similar species.

## 4 Discussion

We contrast the results from the present study with those from the literature in Fig. 7. In the literature studies, lichen samples were broken up by grinding and or homogenising to release ice nucleating entities prior to testing in a freezing assay (Kieft, 1988; Kieft and Ahmadjian, 1989; Kieft and Ruscetti, 1990; Ashworth and Kieft, 1992; Eufemio et al., 2023). In contrast, we placed lichen in water and agitated the samples and then tested the INP content of the resulting solution/suspension without breaking the main body of the lichens up. Hence, it is not a surprise that the concentration of INP per unit mass of original lichen is at the low end of the range defined by the literature data. But, it is striking that even with our much more gentle INP extraction approach than those employed in the literature, still very active INP are released into suspension. It is noteworthy that the earlier studies (from the 1980s and 1990s) quote their blanks as freezing in between $-10\,^\circ$C and $-15\,^\circ$C, while blanks in this study showed background freezing mostly occurred below about $-25\,^\circ$C, allowing measurements at lower temperatures in our study (see Appendix).

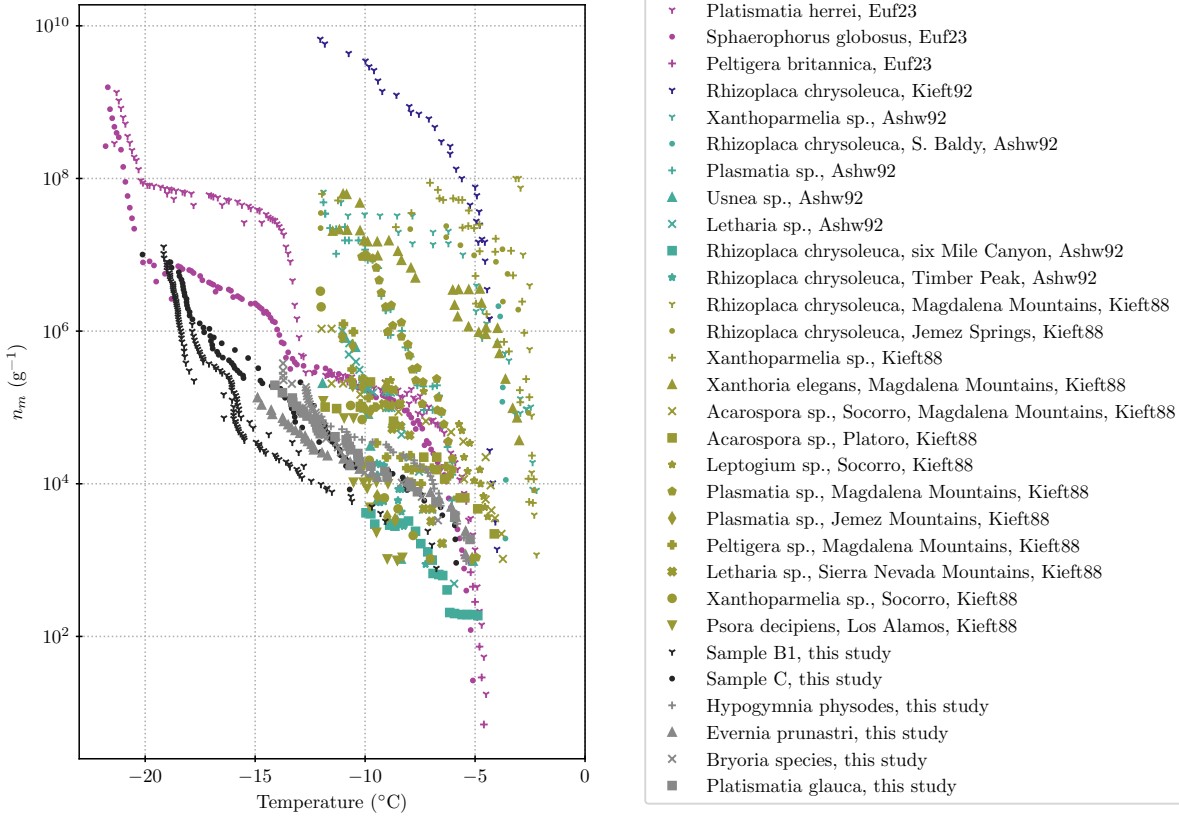

**Figure 7.** Compilation of literature values of INPs per gram of lichen in comparison with values from the present study. Literature data are from Eufemio et al. (2023) as well as Kieft (1988); Ashworth and Kieft (1992); Kieft and Ruscetti (1992).

Lichens in the genus *Platismatia* were tested in the present study and by both Ashworth and Kieft (1992) and Kieft (1988). The samples of *Platismatia sp.* from the Magdalena Mountains in New Mexico had very similar INP content to our sample at around $-5\,^{\circ}$C, with a concentration of $1 \times 10^3$ INPs per gram of lichen. However, the INP concentrations of *Platismatia sp.* from the Magdelana Mountains rise much more rapidly on decreasing temperature, resulting in a concentration of nearly $1 \times 10^8$ INPs per gram of lichen at $-11\,^{\circ}$C, whereas in this study, the Platismatia glauca sample showed a concentration of $1 \times 10^5$ INPs per gram of lichen at $-11\,^{\circ}$C. It is also evident that samples in the same genus have very different activities even when the same techniques are applied. Kieft (1988) show that a *Platismatia* sample from the Jemez Mountains had a much lower activity, with $1 \times 10^6$ INPs per gram at $-11\,^{\circ}$C, much closer to the values reported for our *Platismatia* sample.

Eufemio et al. (2023) very recently presented a study of 29 lichen species from Alaska, some of which were sampled from boreal forests (in addition to the three full spectra reproduced in 7 Eufemio et al. (2023) report $T_{50}$ for additional samples). They also found high variability in ice nucleating activity between species of lichen ($T_{50}$ of -8 and $-15\,^{\circ}$C for the boreal samples; compare to Fig. B1) as well as sensitivity to heat. Their detailed analysis on three lichen species (see 7) demonstrates that there

are two populations of ice active material, one active around $-7\,^\circ\mathrm{C}$ and one at around $-14\,^\circ\mathrm{C}$. They also showed that, while the samples were generally sensitive to heat, these different populations of ice active material responded differently to their heat treatment. They interpreted this as evidence that there are different molecular compositions of ice nucleating materials in lichens.

The size and heat sensitivity of ambient INPs during HyICE-2018 has some consistency with the properties of the lichenaceous INPs we studied here. Schneider et al. (2021) report that the ambient INPs were strongly heat sensitive with all activity above $-13\,^\circ\mathrm{C}$ being removed on heating. The size of INP during HyICE-2018 is also reported by Porter et al. (2020) who revealed that the $0.25\,\mu\mathrm{m}$ to $0.5\,\mu\mathrm{m}$ fraction contained more INPs (above $-22\,^\circ\mathrm{C}$) than any of the larger size fractions in their tests. Porter et al. (2020) comment that the more normal dependency, based on literature data, is that larger aerosol particles
contribute more INPs than smaller aerosol particles, hence their finding was unexpected. Alternatively, the biological INPs observed during HyIce-2018 might have come from a different source. Possibilities include release of INP from the needles or other surfaces of pine trees (Seifried et al., 2023) or perhaps from blowing snow that might release aerosol if snow particles sublime (Frey et al., 2020).

       Kieft and Ruscetti (1990) is the only literature study that looks at the size of the lichenous INPs in droplet assays. Their
samples were filtered through a $0.2\,\mu\mathrm{m}$-pore-size filter. The samples had been centrifuged and only the supernatant was used for testing, whereas in the present study the whole suspension was filtered. Kieft and Ruscetti (1990) mention that their "extraction procedure did not remove all of the nuclei from the lichens" (Kieft and Ruscetti, 1990, p. 3521). This supports the results and the hypothesis brought forth in the present study: As the present study has found the bulk of lichenous INPs to be in between $0.1$ and $2\,\mu\mathrm{m}$ in size, it seems likely that these are not whole spores or diaspores. The very smallest recorded lichen spore is $1\,\mu\mathrm{m}$
in size, and vegetative diaspores are even larger. Hence, as proposed earlier, the INPs might be smaller particles (nano-INPs) or fragments of dispersal particles. As pointed out in O'Sullivan et al. (2015), for example, pollen harbour nano-scale entities that are attached to the pollen grains and are more numerous than the whole pollen grains. Similarly, the INPs found in this study are smaller than whole spores, soredia or isidia. They could become airborne when attached to those larger propagules or wind might pick up these smaller particles by coincidence. Alternatively, the INP may also be bacteria living symbiotically
with the lichen. As mentioned in Sec. 1, the bacteria that Kieft (1988) cultivated from lichen showed no ice-nucleating activity, but not all ice-nucleating bacteria are easily cultured.

       However, when looking at their atmospheric relevance, it is not only important to know the INPs' size fraction, but the structure of the lichen that the particles stem from is bound to be important as well. For example, looking at Fig. 2, one can imagine that wind blowing over these lichens, as a possible way of dispersal for the INPs, interacts with each of them very
differently. For example, *Bryoria sp.* is found to be hanging from trees and sways in the wind. Hence, particles that stem from this lichen species seem likely to be dispersed and lifted into the atmosphere more easily than particles on species that are tightly bound to trunks and branches. As discussed in Sec. 1, propagule disperal mechanisms have been related to meteorological conditions. The lichen tested in the present study were sampled at subzero temperatures (see Fig. 5 in Schneider et al. (2021)). Correlating meteorological variables (temperature, relative humidity and wind speed) with the INP concentrations obtained

from air filter measurements (Brasseur et al., 2022), could shed light onto the hypothesis that lichenous INPs could be a local source in Hyytiälä.

## 5    Conclusions

A possible local source of INPs in Hyytiälä was explored with the thorough examination of the ice-nucleating ability of lichen sampled during the HyICE-2018 campaign. In accordance with literature (Kieft, 1988; Ashworth and Kieft, 1992; Moffett

et al., 2015), the mixtures of three lichen species sampled in Hyytiälä were shown to harbour INP active at temperatures as warm as $-5\,°C$ with concentrations of $1 \times 10^3\,g^{-1}$ of lichen. These findings were expanded through size segregation and heat tests. Many of the lichenous INPs were found to be between $0.1$ and $2\,μm$ in size when immersed in water and those active at temperatures higher than $-18\,°C$ were heat labile.

     As mentioned in the introduction, during HyICE-2018 the forest floor was covered in snow, thus preventing emission of

bioaerosol associated with leaf litter or soil, whereas copious quantities of lichen were exposed to the air. Thus, a viable explanation for the heat sensitivity and the size of ambient atmospheric INPs during HyICE-2018 is that they are derived from lichens.

     Out of the cumulative nucleus spectra, two species of INP could be identified: one species, active at $-16\,°C$, was found to be heat labile and smaller than $0.02\,μm$; the other species was active at $-18\,°C$. The latter's concentration decreased upon filtering

through a $2\,μm$ pore size filter, but it was still detectable in the $0.02\,μm$ size fraction. In the smaller size fractions $< 2\,μm$, this INP species was heat labile, but in the larger size fractions it was not. These differing sensitivities to heat across different size ranges suggest that the INP species responsible for freezing at $-18\,°C$ was present in two different states, attached to a larger particle or free in solution or in different states of aggregation. If it was attached to larger entities or in large aggregates it would be lost on filtration, but is also apparently heat stable. In contrast when it is attached to small particles or in free solution, it is

more sensitive to heat.

     In the species specific experiments, the four species of lichen showed similar ice-nucleating activity, ranging from $1 \times 10^3$ INP per gram of lichen at about $-5\,°C$ to $4 \times 10^5$ INP per gram of lichen at about $-14\,°C$. However, the species harbour differently sized INPs, as the activity decrease seen upon filtration through a $2\,μm$ pore sized filter varied by about one order of magnitude in between species. This implies that some species of lichen may be more important as a source of INPs than

others.

     The size of INPs found in this study suggests that whole spores or soredia are not required for ice nucleating activity, but rather smaller entities nucleate ice. This is analogous to pollen and fungal materials, where nanoscale ice-nucleating entities can become seperate from their host (O'Sullivan et al., 2014). This supports the idea that these INPs could become airborne, either attached to (or part of) spores, soredia or isidia or simply carried aloft by wind by themselves. As the lichen species

investigated here were sampled in Hyytiälä without any knowledge about their ice nucleation activity and all were found to harbour INPs, these findings hold promise (in combination with literature data) that lichen in general represent a source of atmospheric INPs, as they populate many terrestrial environments in great abundance. This source may be especially relevant

in winter when the ground and with it other sources of biological INPs are snow covered in boreal forests. Expanding upon lichen dispersal studies with emphasis on INPs, for example wind tunnel experiments could be employed in combination with an online INP counter such as the Portable Ice Nucleation Experiment (Möhler et al., 2021), controlling environmental factors such as temperature, relative humidity and wind speed. Further it would need to be evaluated if and in which concentrations lichenaceous INPs reach atmospheric heights and locations where they might nucleate ice in clouds.


*Code and data availability.* Processed measurement data and plotting scripts are archived at https://doi.org/10.5281/zenodo.8355809. The data processing for Fig. C1 was done with the HUB code from de Almeida Ribeiro et al. (2023).

 **Appendix A:  Blanks**

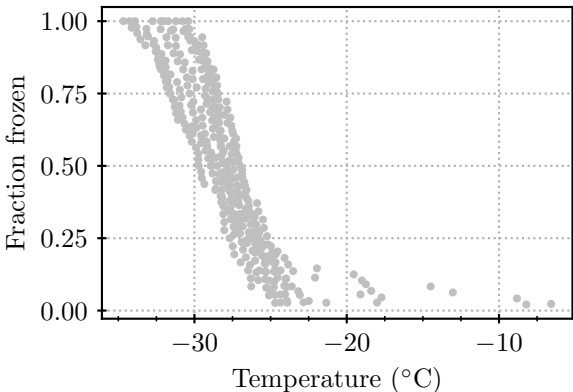

**Figure A1.** Blanks that were run on the morning of each day that experiments were conducted, as described in Sec. 2.2.

**Appendix B:  Comparison of the extraction techniques**

The comparison of sample B1 and C can be seen in Fig. B1. Generally, greater concentrations of INP were present in sample C than in sample B1. As outlined in section 2.2, lichen samples B were mixed by hand for $10\,\mathrm{min}$ and C was mixed on a rotary mixer $30\,\mathrm{min}$. The different procedures might contribute to the greater concentrations of INPs being released with the rotary mixer. This is consistent with the results for B1 and B2 where we saw more INP released with time, indicating sensitivity to the exact experimental procedure.

**Appendix C:  Differential spectra**

The 'steps' discussed in Sec. 3 refer to different populations of ice-nucleating particles. These can be visualized in a differential spectrum computed from $n_m$. We employed the heterogeneous underlying-based (HUB) package from de Almeida Ribeiro et al. (2023) to do this backward calculation. The settings of the script were left to the default ones, except for the number of points for the spline fit set to 50. However, we noticed that in our case the algorithm was sensitive to the random seed and the other assumptions. Thus the results are used here for illustrative purposes only. Fig. C1 shows the results for those spectra, which support the conclusions on 'steps' or different ice-nucleating species drawn in Sec. 3.

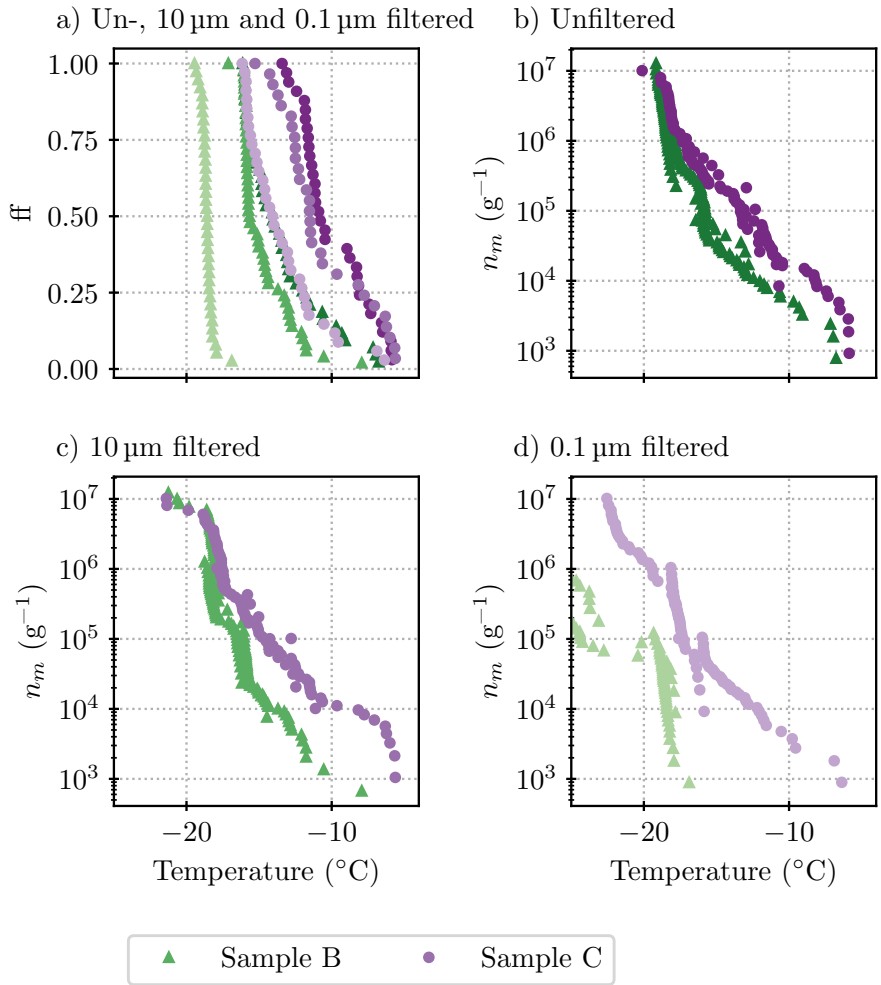

**Figure B1.** Comparison of the filtered size fractions of lichen sample B (green, compare Fig. 5) and C (purple, compare Fig. 6). For better readability no error bars are shown.

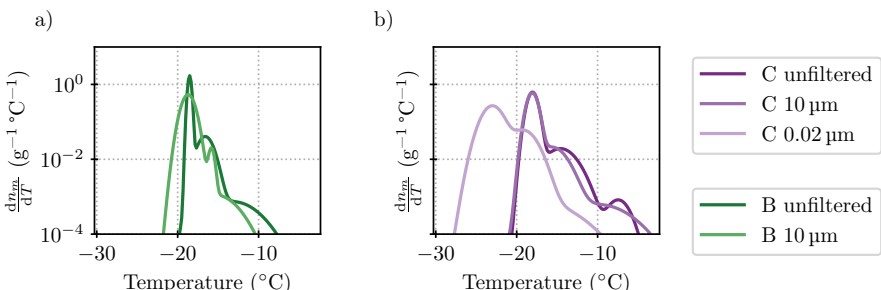

**Figure C1.** Differential spectra for the filtered samples B and C (**a)** and **b)** respectively). Note that other size fractions are not shown because the data from different dilutions did not line up well enough to allow for a sensible interpolation between them. Similarly, for the $10\,\mu m$ size fraction of sample B, the point at $-14\,°C$ and $8.4\times10^4\,g^{-1}$ needed to be excluded to allow for a smooth fit.

*Author contributions.* UP conducted the experiments and analysis and wrote the manuscript. MPA and BJM conceived the idea of the study. MPA, GEP, MH and BJM contributed to the design of the study and the analysis of the results. BJM sampled the lichen. MH and UP took the lichen images. JB identified the lichen species sampled. All authors provided comments and edits to the manuscript.

*Competing interests.* The authors declare no conflict of interest.

*Special issue statement.* This article is part of the special issue "Ice nucleation in the boreal atmosphere".

*Acknowledgements.* We thank the team of the SMEARII station for their efforts during the HyICE-2018 intensive campaign. We are grateful for support from and fruitful discussions with the HyICE-2018 team, in particular Jonathan Duplicy, and the members of the atmospheric ice nucleation group at Leeds University, in particular Mark Tarn and Tom Whale. We thank three anonymous reviewers and Hinrich Grothe for their constructive comments which improved the manuscript substantially.

*Financial support.* This research was supported by the European Research Council (ERC, MarineIce; grant no. 648661), the Engineering and Physical Sciences Research Council (Groovy crystals, EP/M003027/1) and the Natural Environment Research Council (NE/T00648X/1). We are grateful to the EU H2020 ACTRIS-2 for a mobility grant to access the Hyytiälä forestry station as part of the HyIce project (SMR7 RP3 HyICE18, 654109).

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
