# Peer review of "Measurement report: The ice-nucleating activity of lichen sampled in a northern European boreal forest"

_EGUsphere, 2023_

## Author Comment (AC1)

**Author Response to Reviews of**

**Measurement report: The ice-nucleating activity of lichen sampled in a northern European boreal forest**

Ulrike Proske, Michael P. Adams, Grace C. E. Porter, Mark Holden, Jaana Bäck, and Benjamin J. Murray

*ACP,* `doi:10.5194/egusphere-2023-2780`
* * *
RC: *Reviewer Comment*,     AR: *Author Response*,     ☐ Manuscript text

**1.  Reviewer Comment # 1**

AR:  *We thank the reviewer for taking the time to write constructive comments.*

RC:  ***The work by Proske et al. investigates the role of lichen IN from a borreal forest as biological INPs. The authors investigate four different lichen species and state in their introduction that ...While considerable attention has been paid to INPs of bacterial origin, there has been comparably little interest in the ice-nucleating ability of particles that stem from lichen. However, a quick google scholar search revealed that this statement is incorrect. In fact, a study by Eufemio et al. (2023) investigated Lichen as potential INPs in great detail. It is puzzling how the authors have overlooked this study as the topic is identical to their study. In fact, of the four investigated lichen species in this manuscript, three were already reported by Eufemio et al. The samples were also derived from similar snow-covered ecosystems. Eufemio et al. had also already done tests for heat stability and freeze-thaw cycles. In light of that manuscript, the current study does not add any new information to the topic besides confirming the results by Eufemio et al.***

AR:  *We thank the referee for bringing this important recent paper by Eufemio et al. to our attention. We agree that it should be cited in the introduction and our results should be discussed in light of this paper. However, we disagree with the statement that our study 'does not add any new information to the topic'. Eufermio et al. present a survey of 29 lichen species from Alaska and find that the T50 values range from -5.2°C to -14.5°C. They produced these results by taking bulk lichen samples and grinding them, thus releasing ice-nucleating entities from within the body of the lichen as well as those that might reside on the surface and might be more likely to become aerosolised. They also showed that the heat sensitivity is complex, with different species responding in different ways. They did not study the size of the ice-nucleating entities. Based on Eufermio et al. it is not clear at all what we would have expected in our study for lichens collected in southern Finland. Our study is the first to report the ice nucleating activity of lichens from S. Finland's Boreal forests. Furthermore, Hytiala is an important atmospheric research site and it is therefore all the more important to characterise potential sources of INP there. We have added references and discussion of Eufemio throughout the paper. In doing so, we restructured the introduction. We explicitly discuss the Eufemio paper in the following:*

> *Several studies have shown that lichens from a range of environments and across multiple lichen species nucleate ice (Kieft, 1988; Kieft and Ahmadjian, 1989; Ashworth and Kieft, 1992; Moffett et al., 2015; Eufemio et al., 2023). In an early study Kieft (1988) examined 15 lichen. Nearly all of them showed ice-nucleating activity at −8 °C, with −2.3 °C as the highest onset temperature. The bacteria that could be cultivated from the lichen showed no ice nucleation activity. Moffett et al. (2015) and Eufemio et al. (2023) between them surveyed the ice nucleation activity of 86 lichen samples and found that while ice nucleation was ubiquitous these lichens had remarkably varied ice nucleating abilities. Moffett et al. (2015) report onset freezing ranging from −5.1 °C to −20 °C, while Eufemio et al. (2023) report median freezing temperatures between −5.2 °C to −14.5 °C. In addition, there is substantial variability in ice nucleation between different samples of the same species of lichen. For example, one sample of Evernia Prunastri nucleated ice at −5.6 °C while another nucleated ice at −10 °C (Moffett et al., 2015). These studies show the ubiquity of ice nucleation in lichens, but given the observed variability in ice nucleating activity, we cannot simply infer that lichens in one environment posses the same ice nucleating activity as the same lichen species in other environments.*

*And in the discussion we include:*

> *Eufemio et al. (2023) recently presented a study of 29 lichen species from Alaska, some of which were sampled from boreal forests. They also found high variability in ice nucleating activity between species of lichen as well as sensitivity to heat. They performed detailed analysis on three lichen species demonstrating that there are two populations of ice active material, one active around −7 °C and one at around −14 °C. They also showed that, while the samples were generally sensitive to heat, these different populations of ice active material responded differently to their heat treatment. They interpreted this as evidence that there are different molecular compositions of ice nucleating materials in lichens.*

*We have also improved the conclusions section, including adding a short paragraph on how our measurements are consistent with measurements of ambient INP during HyICE-2018.*

> *The size and heat sensitivity of ambient INPs during HyICE-2018 has some consistency with the properties of the lichenaceous INPs we studied here. Schneider et al. (2021) report that the ambient INPs were strongly heat sensitive with all activity above −13 °C being removed on heating. The size of INP during HyICE-2018 is also reported by Porter et al. (2020) who revealed that the 0.25 µm to 0.5 µm fraction contained more INPs (above −22 °C) than any of the larger size fractions in their tests. Porter et al. (2020) comment that the more normal dependency, based on literature data, is that larger aerosol particles contribute more INPs than smaller aerosol particles, hence their finding was unexpected. As mentioned in the introduction, during HyICE-2018 the forest floor was covered in snow, thus preventing emission of bioaerosol associated with leaf litter or soil, whereas copious quantities of lichen were exposed to the air. Thus, a viable explanation for the heat sensitivity and the size of ambient atmospheric INPs during HyICE-2018 is that they are derived from lichens.*

**RC:** *In addition, the current work has additional flawes in their manuscript that should be addressed. The study puts emphasis on the potential size dependency of the lichen INP. Yet again an important study on this topic seem to have been overlooked as Kieft and Ruscetti (1992) used gamma radiation to determine size of the INs.*

AR: *We have added a sentence on this paper in the introduction:*

> *Using a relationship between molecular size and the likelihood to become deactivated on exposure to gamma radiation, Kieft and Ruscetti (1992) found a logarithmic relationship between freezing temperature and protein size.*

RC: **How did the authors determined which lichen species they investigated? visual inspection? genetic testing?**

AR: *We used a visual inspection method, similar to Eufemio et al., in combination with the knowledge of what common lichen species are present in Southern Finland. While DNA analysis would provide robust data, in a 'normal' production forest in a boreal climate zone, the possible number of species is rather modest, so visual assessment is sufficient. The species listed here are very typical to this kind of forest.*

RC: **How did the authors ensure that the lichen were not contaminated with e.g. bacteria on top of the lichen. Any washing steps prior to analysis?**

AR: *We took the approach where we did not wash the samples, unlike in Eufemio et al. paper. Washing would likely remove ice nucleating material that is loosely bound to the surface of lichen. We think that it is this loosely bound material, such as the soredia or isida (that can become aerosolised) that is likely to be important in the atmosphere. Furthermore, washing might remove bacteria that are now thought to be part of the symbiosis in lichen (Grimm et al., 2021). Hence, we chose not to wash the samples prior to use. We have added a new paragraph in the methods section to make our rational clear.*

> Examples of the structures that can become aerosolised are shown in Fig. 3. Panel a shows soredia on *Platismatia glauca*, while panel b shows isidia on *Evernia prunastri*. These vegetative diaspores can be broken off the thallus through the action of wind, rain droplets or even animals. The recognition that it is likely these fragile structures on the surface of the lichen that preferentially become airborne helped us to design a droplet freezing assay that is appropriate. In some previous ice nucleation studies the lichens were ground with some water to produce a pulp that was then suspended in water (Kieft, 1988; Eufemio et al., 2023). This approach might be appropriate for studying the water harvesting properties of lichens, but may be less relevant for understanding atmospheric implications. In addition, the practice of washing lichen samples to remove non-lichen ice-nucleating entities may inadvertently remove the soredia and isidia, the very entities of particular interest. Hence, we used an approach where lichen was exposed to water and gently agitated in order that fragile structures, like the soredia or isidia, might be removed (details in the next section). The large pieces of lichen were allowed to settle to the bottom of the vial and then the aqueous supernatant, which was clear to the eye, was sampled for the droplet freezing assay. While this approach does not replicate bioaerosol production from lichen, it does bias the analysis towards the entities associated with lichen that are likely to become aerosolised.

RC: **Lichen is a symbiosis of more then two partners, the idea that is only two is outdated and has been disproven.**

AR: *Thank you for pointing us to this relatively new knowledge. We had focused on the text-book understanding, which is obviously somewhat out of date. We have added the concept to the text as follows:*

> For many years lichens were though to be symbiotic organisms composed of a fungal partner, the mycobiont, and a photobiont partner (Nash, 2008). However, it is now recognised that in addition to the mycobiont and photobiont (algae/cyanobacteria), lichen species can accommodate several additional symbionts, including yeasts and bacteria, associated with the fungus or locally living in the microhabitats of lichen thalli (Aschenbrenner et al., 2016; Cernava et al., 2017; Grimm et al., 2021).

**RC:** *p.3 l. 76, the Reference Schwidetzky et al. (2023a) would be fitting as it is the first study that provides conclusive evidence that fungal IN are proteins*

**AR:** *We added this citation:*

> *We also know that some fungal materials produce proteins that nucleate ice effectively and these proteins can become separated from the mycelia (O'Sullivan et al., 2015; Schwidetzky et al., 2023b).*

*and*

> *This size dependence is consistent with the idea that larger aggregates of proteins have the potential to nucleate ice at higher temperatures (Schwidetzky et al., 2023a).*

**References**

Aschenbrenner, Ines A., Tomislav Cernava, Gabriele Berg, and Martin Grube (Feb. 2016). "Understanding Microbial Multi-Species Symbioses". In: *Frontiers in Microbiology* 7. ISSN: 1664-302X. DOI: 10.3389/fmicb.2016.00180.

Ashworth, Edward N. and Thomas L. Kieft (1992). "Measurement of Ice Nucleation in Lichens Using Thermal Analysis". In: *Cryobiology* 29, pp. 400–406.

Cernava, Tomislav, Armin Erlacher, Ines Aline Aschenbrenner, Lisa Krug, Christian Lassek, Katharina Riedel, Martin Grube, and Gabriele Berg (July 2017). "Deciphering Functional Diversification within the Lichen Microbiota by Meta-Omics". In: *Microbiome* 5.1, p. 82. ISSN: 2049-2618. DOI: 10.1186/s40168-017-0303-5.

Eufemio, Rosemary J, Ingrid de Almeida Ribeiro, Todd L Sformo, Gary A Laursen, Valeria Molinero, Janine Fröhlich-Nowoisky, Mischa Bonn, and Konrad Meister (2023). "Lichen Species across Alaska Produce Highly Active and Stable Ice Nucleators". In: *Biogeosciences* 20. DOI: 10.5194/bg-20-2805-2023.

Grimm, Maria, Martin Grube, Ulf Schiefelbein, Daniela Zühlke, Jörg Bernhardt, and Katharina Riedel (Mar. 2021). "The Lichens' Microbiota, Still a Mystery?" In: *Frontiers in Microbiology* 12. ISSN: 1664-302X. DOI: 10.3389/fmicb.2021.623839.

Kieft, Thomas L (1988). "Ice Nucleation Activity in Lichens". In: *Applied and Environmental Microbiology* 54, p. 5.

Kieft, Thomas L. and Vernon Ahmadjian (Oct. 1989). "Biological Ice Nucleation Activity in Lichen Mycobionts and Photobionts". In: *The Lichenologist* 21.04, pp. 355–362. ISSN: 0024-2829, 1096-1135. DOI: 10.1017/S0024282989000599.

Kieft, Thomas L. and Tracy Ruscetti (June 1992). "Molecular Sizes of Lichen Ice Nucleation Sites Determined by Gamma Radiation Inactivation Analysis". In: *Cryobiology* 29.3, pp. 407–413. ISSN: 00112240. DOI: `10.1016/0011-2240(92)90042-Z`.

Moffett, B. F., G. Getti, S. K. Henderson-Begg, and T. C. J. Hill (Jan. 2015). "Ubiquity of Ice Nucleation in Lichen — Possible Atmospheric Implications". In: *Lindbergia* 3.1, pp. 39–43. ISSN: 0105-0761. DOI: `10.25227/linbg.01070`.

Nash, Thomas H. (2008). *Lichen Biology.* Leiden: Cambridge University Press. ISBN: 978-0-511-41407-7.

O'Sullivan, D., B. J. Murray, J. F. Ross, T. F. Whale, H. C. Price, J. D. Atkinson, N. S. Umo, and M. E. Webb (July 2015). "The Relevance of Nanoscale Biological Fragments for Ice Nucleation in Clouds". In: *Scientific Reports* 5.1, pp. 1–7. ISSN: 2045-2322. DOI: `10.1038/srep08082`.

Porter, Grace C. E., Sebastien N. F. Sikora, Michael P. Adams, Ulrike Proske, Alexander D. Harrison, Mark D. Tarn, Ian M. Brooks, and Benjamin J. Murray (June 2020). "Resolving the Size of Ice-Nucleating Particles with a Balloon Deployable Aerosol Sampler: The SHARK". In: *Atmospheric Measurement Techniques* 13.6, pp. 2905–2921. ISSN: 1867-8548. DOI: `10.5194/amt-13-2905-2020`.

Schneider, Julia et al. (Mar. 2021). "The Seasonal Cycle of Ice-Nucleating Particles Linked to the Abundance of Biogenic Aerosol in Boreal Forests". In: *Atmospheric Chemistry and Physics* 21.5, pp. 3899–3918. ISSN: 1680-7324. DOI: `10.5194/acp-21-3899-2021`.

Schwidetzky, Ralph et al. (2023a). "Functional Aggregation of Cell-Free Proteins Enables Fungal Ice Nucleation". In: *Proceedings of the National Academy of Sciences* 120.46.

Schwidetzky, Ralph et al. (2023b). "Functional aggregation of cell-free proteins enables fungal ice nucleation". In: *Proceedings of the National Academy of Sciences* 120.46, e2303243120. DOI: `10.1073/pnas.2303243120`. eprint: `https://www.pnas.org/doi/pdf/10.1073/pnas.2303243120`. URL: `https://www.pnas.org/doi/abs/10.1073/pnas.2303243120`.

This document was generated with a layout template provided by Martin Schrön (`github.com/mschroen/review_response_letter`).

---

## Author Comment (AC2)

**Author Response to Reviews of**

**Measurement report: The ice-nucleating activity of lichen sampled in a northern European boreal forest**

Ulrike Proske, Michael P. Adams, Grace C. E. Porter, Mark Holden, Jaana Bäck, and Benjamin J. Murray
*ACP,* `doi:10.5194/egusphere-2023-2780`

RC: *Reviewer Comment*,    AR: *Author Response*,    ☐ Manuscript text

**1.  Reviewer Comment # 2**

AR:  *We thank the reviewers for taking the time to write constructive comments.*

RC:  **The manuscript presents an analysis of the ice forming activity of lichen wash water. Lichen samples growing on Scots pines in Hyytiäla, Finland, were collected, washed, and the ice forming activity of the water was determined using a drop freezing experiment. In addition, the washing water was filtered and subjected to heat-treatment to study the size and nature of the ice nucleating particles.**

RC:  **The manuscript is well written and gives an overview of many previous studies. However, the new experimental work does not appear to go beyond the screening a few samples and the interpretation of results appears speculative. In its current form the manuscript does not present substantial and convincing new data.**

AR:  *This is a study that is explicitly focused on lichens in Hyytiäla in light of the measurements that show a substantial population of biological INP from an unknown source. It is not intended to go beyond the common lichens that were present in the forest during the HyICE campaign. Importantly, no one has studied the lichens present in S. Finland in the past and given the massive variability observed in other locations in ice nucleating activity (even within single species) it was not clear to what extent the lichens in Hyytiälä would nucleate ice. Furthermore, many previous studies have taken entire pieces of lichen or ground lichen samples to release ice nucleating entities into water. We have take a more subtle approach of washing the lichens in an attempt to release only those entities that might become aerosolised. We have successfully addressed the question whether lichens in Hyytiälä contain ice nucleating material that could conceivably become aerosolised. This shows that it is worth the time and effort involved in performing future aerosolisation work where someone might attempt to quantify the production of INPs from lichen as a function of wind speed, RH etc. In order to make our objectives clearer, we have restructured the introduction and included the following at the end of the introduction:*

*In this paper we report the ice nucleating ability of lichens that were present and exposed in the boreal forest at the Station for Measuring Ecosystem-Atmosphere Relations (SMEAR) II located in Hyytiälä, Finland during the HyICE-2018 campaign. The HyICE-2018 campaign was focused on measuring atmospheric INPs between February and June 2018 (Brasseur et al., 2022). Schneider et al. (2021) report the presence of heat sensitive biological INPs during the campaign. However, the source of these INPs was unclear since the surface was snow covered, which rules out leaf litter or bare soil as sources of INPs. We hypothesis that the trees which bore lichen, that were exposed even in the winter when the canopy and the ground were snow covered (see Fig. 1), might have been a source of INPs. In this paper we tackle the question of if the lichens in the forest during HyICE-2018 contained ice-nucleating entities. We see this as a first step to addressing our hypothesis and a positive outcome would provide motivation to address the question of whether sufficient quantities of INPs are released into the atmosphere to influence the INP population and subsequent cloud formation.*

*In order to address the referees concerns around speculation, we revisited our abstract and focused it. The modified section is below:*

*INPs derived from lichen sampled during HyICE-2018 are shown to nucleate ice at temperatures as warm as $-5\,°C$ in aqueous suspensions extracted from $0.03\,\mathrm{g\,mL^{-1}}$ lichen. Successive filtration to smaller sizes removes some of the most active INPs in suspension, but substantial activity remains even when filtering to $0.02\,\mu m$. The small size of the INPs from lichen means they have the potential to either be emitted directly into the atmosphere or be associated with larger particles, such as lichenous reproductive aerosol types (spores, or diaspores). We also show that the INPs from lichens from Hyytiäläre sensitive to heat, which is similar to the INP sampled from the atmosphere of Hyytiälä and consistent with the presence of ice-active proteins. This study shows that lichen from a European Boreal forest in Hyytiälä harbour INPs, which may be especially important in this snow covered habitat where few, if any, other biological INP sources are available. The great terrestrial abundance of lichens in Hyytiälä, and around the world, calls for further research to combine their ice nucleating ability with dispersal studies to evaluate the flux of lichenous INPs into the atmosphere as well as to what extent these particles reach heights and locations where they might influence cloud properties.*

**1.1. Introduction**

**RC:** *Line 79-80: Explain how ice formation might help to change physiological activity and indicate in what way.*

AR: *We have edited this section:*

Once a small amount of water is frozen on the thallus, more water may preferentially deposit on it. Later, when the temperature increases, this ice may melt and the liquid water would become available to the lichen. This process is all the more important, since lichen lack stomata and are therefore not able to actively control water loss as many plants do (Kappen and Valladares, 2007).

**1.2. Methods**

**RC:** *Line 90 ff: Clarify why the approach was to use a mixture of lichen species for heat-treatment and detailed filtering experiments rather than a single species. The approach seems more suitable for an initial*

*screening or a feasibility study.*

AR: *We wanted to know if the lichens present in Hyytiäla contained ice nucleating material. Knowledge of the ice nucleating ability of the single species is clearly academically interesting, but the first order question is if the lichens in Hyytiäla contain ice nucleating material. We have made this clearer:*

> *We examined mixed samples of lichen for their ice nucleating ability, size of the ice-nucleating species and the heat sensitivity rather than solely focusing on single species in order to reveal if the lichens in Hyytiäla harboured ice nucleating entities and obtain an indication of their activity. This allowed us to address our stated objective of determining if there is a potential source of biological INPs associated with the prolific lichen population in Hyytiäla.*

RC: **Line 110 ff. Was the lichen separated from the tree bark?**

AR: *Yes, it was. This is now stated in the methods.*

RC: **Line 114: Explain why an attempt was made to mimic sample B.**

AR: *We wanted roughly the same proportions of the lichen species to get a replica and to see how different the results would be if we changed the method of agitation in water. Sample B was carefully rotated in water by hand for 10 mins, while C was mixed on a rotary mixed for 30 mins at 30 rotations per min. These details have been added to the table as well as the text. In the text we have clarified this and now state:*

> *Both sample B and C had a similar proportion of the different lichen species, but the sensitivity to the mixing method was explored.*

RC: **Line 118 ff., 123ff: Explain why the particle fraction resembling the windblown particles is suspended (or not) in a gentle, wet extraction. What difference can be expected between wind and water extraction? Smaller or larger particles, more or less numerous? With the aim of using this type of measurement to estimate the atmospheric ice formation potential of lichens from the lichen mass, the similarity of particles washed off to wind-blown needs to be clarified.**

AR: *At no point have we claimed that the wet extraction processes produces a 'particle fraction resembling the windblown particles'. But, we do think our approach is more relevant to the atmosphere than simply grinding whole lichen samples. We have added the following text to the methods to explain our rational:*

Examples of the structures that can become aerosolised are shown in Fig. 3. Panel a shows soredia on *Platismatia glauca*, while panel b shows isidia on *Evernia prunastri*. These vegetative diaspores can be broken off the thallus through the action of wind, rain droplets or even animals. The recognition that it is likely these fragile structures on the surface of the lichen that preferentially become airborne helped us to design a droplet freezing assay that is appropriate. In some previous ice nucleation studies the lichens were ground with some water to produce a pulp that was then suspended in water (Kieft, 1988; Eufemio et al., 2023). This approach might be appropriate for studying the water harvesting properties of lichens, but may be less relevant for understanding atmospheric implications. In addition, the practice of washing lichen samples to remove non-lichen ice-nucleating entities may inadvertently remove the soredia and isidia, the very entities of particular interest. Hence, we used an approach where lichen was exposed to water and gently agitated in order that fragile structures, like the soredia or isidia, might be removed (details in the next section). The large pieces of lichen were allowed to settle to the bottom of the vial and then the aqueous supernatant, which was clear to the eye, was sampled for the droplet freezing assay. While this approach does not replicate bioaerosol production from lichen, it does bias the analysis towards the entities associated with lichen that are likely to become aerosolised.

RC: *Does exposure of lichens to Milli-Q water cause lysis?*

AR: *This is an interesting question that we haven't been able to find a clear answer to.*

RC: *Section 2.4. Clarify whether samples are first filtered and then heated or vice versa.*

AR: *This became a little confusing because we did tests where we explored the effect of the order of treatments. But, in those experiments we also had issues with the reproducible associated with poor mixing. We have decided to remove the dilution series of the heated samples. The undiluted heat tests demonstrate the heat sensitivity and by removing the dilutions we remove complexity and ambiguity (see below). We also only did the dilutions for a subset of the heated samples.*

RC: *Section 2.5.: For brevity, consider referring to previous descriptions and only mention steps specific to the present experiments.*

AR: *We feel that the description of the technique is already concise and specific to the present experiments. It isn't clear what we would remove.*

RC: *Line 162 ff.: If I understand correctly, the initially added lichen mass did not dissolve in the water. Why was the initially added lichen mass used to normalise the measurements if the lichen mass was not contained in the droplets? Also, filtering removes the lichen mass from the suspension. Justify the use of the initially added lichen mass to normalize data from filtered samples.*

AR: *We normalise to the initial mass of lichen because we need to be able to compare results from the various heat and filtration tests. By normalising to initial mass of lichen, changes in $n_m$ reveal relative changes in activity. This is a common approach and understood by the community (e.g. Murray et al. (2012)). We have added:*

We normalise to the initial mass of lichen in order that we can quantify the relative changes in activity on dilution, heat tests and filtration.

**1.3.    Results**

**RC:**  *Line 171: Clarify how the hypothesis can be confirmed by the present study? Considering that the water samples may not be representative of aerosolized samples (line 124-125) and that both the concentration of lichen particles in the atmosphere and their ice forming capacity need to be known to make this assessment (line 83-84).*

**AR:**  *We made the scope of our work much clearer in the final paragraph of the introduction. We agree that more work is needed, but our study shows that the next phase of this work is indeed justified and worthwhile.*

**RC:**  *The measurement data can possibly be better interpreted if they are presented as fraction frozen instead of as site density normalized to lichen mass. Below is a comparison of the fraction frozen of non-heat-treated, non-diluted samples B and C (data taken from the provided Assets to the manuscript).*

[Figure]

Figure 1

**AR:**  *Fraction frozen curves are fine for comparing like-for-like experiments. But, when the samples are diluted or if different concentrations of lichen are used, then fraction frozen curves cannot be compared. The community has moved away from showing fraction frozen curves (e.g. Murray et al. (2012)). However, we do take on board the more general principle that Fig 6 was hard to understand as there was just too much data in it. We have produced a revised fig 6, with multiple panels.*

**RC:**  *Line 177-178: As can be seen in the plot above, there is a clear offset in the fraction frozen (FF) between the unfiltered and the 10μm filtered B sample (blue triangles). They cannot be considered similar.*

AR:    *We have qualified what we mean by similar by adding in 'within $2\,°C$'.*

RC:    **Line 193: The experimental detection limit makes the comparison of heat-lability at different temperatures questionable. There is no data above -15°C in the heat-treated data that can be compared.**

AR:    *On the contrary, the heat test is very clear. We direct the reader to Daily et al. (2022). But, there are clear shifts in the spectra on heating, with no observed freezing above -16°C after heating.*

RC:    **As shown in your Fig. 4 c) and d), there seems to be a large offset between original and dilutions of heat-treated samples. Can this be explained?**

AR:    *There are some offsets, yes. We had done a bit more work on this aspect, but removed it from the paper for the sake of simplicity. We found that there are sometimes issues around pippetting these samples, where if the suspension isn't well mixed you can get more or less material in the dilution than expected. We took these learnings to improve our approach. Given the ambiguity in the dilutions of the heat test we have removed these results from the figure. This does not change the conclusions, but removes ambiguity and complexity.*

RC:    **Line 220: Explain why lower concentrations can be expected for smaller size fractions.**

AR:    *If you remove the higher temperature INP, you naturally reduce the concentration across the full spectrum given $n_m$ is a cumulative quantity. We have added:*

> (because higher temperature INP are removed by filtration and this reduces the concentration across the full spectrum since this is a cumulative quantity)

RC:    **Line 224: The differences between sample B and C (see figure above) are surprisingly large considering all single lichen samples showed similar spectra.**

AR:    *Yes, we think this is attributed to the different extraction method. We have improved the pertinent paragraph which now reads:*

> The comparison of sample B1 and C can be seen in Fig. 6. Generally, greater concentrations of INP were present in sample C than in sample B1. As outlined in section 2.2, lichen samples B were mixed by hand for $10\,min$ and C was mixed on a rotary mixer $30\,min$. The different procedures might contribute to the greater concentrations of INPs being released with the rotary mixer. This is consistent with the results for B1 and B2 where we saw more INP released with time, indicating sensitivity to the exact experimental procedure. It also should be noted that the experimental procedure for estimating the composition of the samples to be the same by eyesight was rather crude. Hence, different species of lichen may have different ice-nucleating characteristics. To explore this further we attempted to separate out the lichen species and test them individually.

RC:    **Ideally, the single species experiments should reproduce the results of the mixed sample when their spectra are added together, weighted according to their proportion in the mixed sample. Can the results of the single species experiments be used to deduce what the difference between B and C might have been?**

AR:    *With standardised processes this might be possible, but to what end? The objective here was to test if the samples contained INPs.*

RC:    **Line 225 ff: Clarify how "more" and "less" heat-labile particles can be explained? Shouldn't "all" or "none" entities that are ice active at a certain temperature be affected?**

AR: *There are degrees of heat sensitivity. This is covered in detail by Daily et al. (2022), who test the heat test. Even with psuedomonas Syringae only a fraction of the ice-nucleating proteins are destroyed and the different classes of protein aggregates are affected to different extents.*

RC: **Fig. 5: Why were no dilutions measured for heat-treated samples C?**

AR: *This is now a mute point given we have removed the heat treated dilutions from B due to issues with reproducibility.*

RC: **Line 231-232: Unclear where this step can be seen. There is no data for the heat-treated samples C, unfiltered and 10µm at -18°C and a step can also be seen in Fig.5e) for 0.1µm.**

AR: *This is much clearer in the revised Fig 6.*

RC: **Line 232: Define which "characteristics for different sizes" are referred to and clarify how this can be used to infer the state of these INPs.**

AR: *We have made this clearer, by inserting '(activity and heat sensitivity)'.*

RC: **Line 238: The logarithmic scale used in Fig. 6 could be misleading here. Overlap can be checked more directly in FF plots. The figure above shows a comparison of the FF of sample B and C. It shows that the freezing signal of the two samples differs considerably.**

AR: *We have revised Fig. 6 to make it clearer.*

RC: **It is striking that sample B, unfiltered produced the same FF as sample C, filtered through 0.1 µm, raising overall doubt on the presented interpretation of size dependence. The data should be replotted as FF and features discussed based on such figures. The nm -plots obscure the data by scaling and plotting on a logarithmic ordinate.**

AR: *As mentioned above, this difference is attributed to the different extraction methods. It is not possible to plot the data in a meaningful manner as fraction frozen plots since the data contains dilutions.*

RC: **Line 240: What is considered 1 order of magnitude (small difference) in Fig. 6 corresponds to about 4°C (large difference) in the FF plot above.**

AR: *We have now qualified the shift in temperature as well as concentration in appropriate places in the text.*

RC: **Line 253: Indicate which small particles.**

AR: *We agree that this isn't clear. Revised to: 'simply show the same INP concentrations because entities such as the soredia and isidia may have been spread throughout the bag'.*

RC: **Line 257: Given the difficulty of reproducing measurements from sample B and C, if seems highly uncertain whether the result of a single measurement is reliable. All measurement should be repeated several times before they are compared and interpreted. Since droplet freezing experiments are neither time consuming nor expensive, several repetitions of experiments are desirable before conclusions are drawn.**

AR: *There is a discrepancy between sample B and C, because the preparation method was different. The discussion mentioned above is now clearer. In an ideal world we would repeat everything in triplicate, but in practice compromises need to be made due to limitations of resources and time. Rather than perform each measurement in triplicate we chose to perform dilutions, which also give a good idea of reproducibility in the region of overlap, but have the added benefit of extending the dataset to lower temperatures.*

**1.4. Discussion**

**RC:** *Line 262: Can recommendations be given on how to measure the atmospherically relevant ice formation activity of lichens in a comparable way?*

AR: *This is something we have been giving thought to and had planned to do. However, the pandemic prevented us from doing this work. At the end of the conclusions section we briefly mention the idea of making use of a wind tunnel and that we could use an online INP counter to quantify INP production as a function of RH, T etc. This would clearly require substantial resources and is beyond the scope of what we can do for this paper.*

**RC:** *Line 264-278: A comparison of "INPs per gram of lichen" between studies in which the lichen material remained in droplets for freezing experiments (ground powder) and experiments with washing water in which the lichen matter is not contained in the water droplets seems incorrect. Explain how this comparison of concentrations can be justified. Consider limiting the comparison to temperatures at which specific features are observed.*

AR: *We prefaced this section with a statement that comparison between different approaches is difficult for exactly this reason. We have emphasises this and now state 'This makes a quantitative comparison of INP concentrations challenging and therefore needs to be done with some caution.'*

**RC:** *Line 290 ff: The reported size of the lichen INPs is questionable as the same freezing curve was measured from sample B, unfiltered and sample C, filtered through 0.1μm filter (see figure above).*

AR: *We dealt with this misunderstanding above.*

**RC:** *Line 304: What is the size of the propagules? Consider adding a definition.*

AR: *This refers to spores, soredia and isidia etc. We defined sizes in the introduction. We have added '(e.g. spores, soredia and isidia)'*

**RC:** *Line 318 f: A correlation to which meteorological variable would provide a strong indication of lichen INP?*

AR: *We have added in brackets the variables we are referring to.*

**RC:** *Line 327: The decrease in concentration with size indicates that a large fraction of these INPs are larger. Clarify what fraction of the -16°C species is smaller than 0.02μm.*

AR: *We would rather not quantify the fraction, this would serve no obvious purpose and be prone to uncertainty.*

**RC:** *Line 329: This conclusion does not appear to be supported by the data. Figure 4 shows a reduction in all sized after heat treatment.*

AR: *The point is these INP are removed only when also filtered to small sizes.*

**RC:** *Line 330: Clarify which characteristics are referred to and explain how they support the suggestion of these two different states.*

AR: *We have attempted to clarify our thinking here:*

> These differing sensitivities to heat across different size ranges suggest that the INP species responsible for freezing at $-18\,°C$ was present in two different states, attached to a larger particle or free in solution or in different states of aggregation. If it was attached to larger entities or in large aggregates it would be lost on filtration, but is also apparently heat stable. In contrast when it is attached to small particles or in free solution, it is more sensitive to heat.

**RC:** *Line 335-336: Clarify how it can be concluded whether a species is an important source of INP based on the current results without knowing the abundance of lichen particles in the atmosphere.*

AR: *We have replace 'are' with 'may be'.*

**RC:** *Line 337ff: Since there is no information on the abundance of such smaller lichen entities in the atmosphere, it is speculation whether they contribute INP. Furthermore, the fact that they may be dissolved in water does not necessarily indicate their presence in the atmosphere.*

AR: *Yes, that is correct. That is why we need wind tunnel measurements.*

**RC:** *Line 340ff: As the authors explain below (Line 343ff), the ice activity of lichens without atmospheric concentration data is not evidence that they are a source of atmospheric INPs, and other approaches are necessary to clarify the importance of lichens as INPs.*

AR: *This is correct and consistent with what we say.*

**1.5. Technical corrections**

**RC:** *Line 18: Provide a reference to "formation of ice in clouds is amongst one of the least well understood of these processes."*

AR: *Done. (Murray et al., 2021; Tan et al., 2016)*

**RC:** *Line 29-30: References should be in brackets.*

AR: *This has been corrected.*

**RC:** *Line 47: should it be "Fruticose" instead of "Fructiose"?*

AR: *This is correct and this has been corrected.*

**RC:** *Line 123: Clarify what is meant by "metal housing". The filter holder?*

AR: *Replaced with 'Advantec 301000 stainless steel filter holder'*

**RC:** *Line 158: Define "EF600"*

AR: *This is the machine's name. Added '(EF600 Stirling engine chiller, Grant-Asymptote)'*

**RC:** *Line 176: consider introducing sample B1 and B2 in Sect. 2.2*

AR: *Added 'Sample B is split into B1 and B2 in the manuscript, B2 was sampled from the same suspension one day later, so had had more time to release INP into suspension/solution. '*

**RC:** *Line 280: ... temperatures in our study.*

AR: *Corrected.*

**RC:** *Line 290: Provide a reference for the smallest lichen spore size of 1μm.*

 AR: *This is defined in some detail in the introduction.*

**References**

Brasseur, Zoé et al. (2022). "Measurement Report: Introduction to the HyICE-2018 Campaign for Measurements of Ice-Nucleating Particles and Instrument Inter-Comparison in the Hyytiälä Boreal Forest". In: *Atmos. Chem. Phys.*, p. 29.

Daily, Martin I, Mark D Tarn, Thomas F Whale, and Benjamin J Murray (2022). "An Evaluation of the Heat Test for the Ice-Nucleating Ability of Minerals and Biological Material". In: *Atmos. Meas. Tech.*, p. 31.

Eufemio, Rosemary J, Ingrid de Almeida Ribeiro, Todd L Sformo, Gary A Laursen, Valeria Molinero, Janine Fröhlich-Nowoisky, Mischa Bonn, and Konrad Meister (2023). "Lichen Species across Alaska Produce Highly Active and Stable Ice Nucleators". In: *Biogeosciences* 20. DOI: `10.5194/bg-20-2805-2023`.

Kappen, L. and F. Valladares (2007). "Opportunistic Growth and Desiccation Tolerance: The Ecological Success of Poikilohydrous Autotrophs". In: *Handbook of Functional Plant Ecology*. Ed. by F. I. Pugnaire and F. Valladares. Boca Raton: CRC Press, pp. 7–65.

Kieft, Thomas L (1988). "Ice Nucleation Activity in Lichens". In: *Applied and Environmental Microbiology* 54, p. 5.

Murray, B. J., D. O'Sullivan, J. D. Atkinson, and M. E. Webb (2012). "Ice Nucleation by Particles Immersed in Supercooled Cloud Droplets". In: *Chemical Society Reviews* 41.19, p. 6519. ISSN: 0306-0012, 1460-4744. DOI: `10.1039/c2cs35200a`.

Murray, Benjamin J., Kenneth S. Carslaw, and Paul R. Field (Jan. 2021). "Opinion: Cloud-Phase Climate Feedback and the Importance of Ice-Nucleating Particles". In: *Atmospheric Chemistry and Physics* 21.2, pp. 665–679. ISSN: 1680-7324. DOI: `10.5194/acp-21-665-2021`.

Schneider, Julia et al. (Mar. 2021). "The Seasonal Cycle of Ice-Nucleating Particles Linked to the Abundance of Biogenic Aerosol in Boreal Forests". In: *Atmospheric Chemistry and Physics* 21.5, pp. 3899–3918. ISSN: 1680-7324. DOI: `10.5194/acp-21-3899-2021`.

Tan, I., T. Storelvmo, and M. D. Zelinka (Apr. 2016). "Observational Constraints on Mixed-Phase Clouds Imply Higher Climate Sensitivity". In: *Science* 352.6282, pp. 224–227. ISSN: 0036-8075, 1095-9203. DOI: `10.1126/science.aad5300`.

This document was generated with a layout template provided by Martin Schrön (`github.com/mschroen/review_response_letter`).

---

## Author Response (AR2)

**Author Response to Reviews of**

**Measurement report: The ice-nucleating activity of lichen sampled in a northern European boreal forest**

Ulrike Proske, Michael P. Adams, Grace C. E. Porter, Mark Holden, Jaana Bäck, and Benjamin J. Murray

*ACP,* `doi:10.5194/egusphere-2023-2780`
* * *
RC: *Reviewer Comment*,    AR: *Author Response*,    ☐ Manuscript text

We thanks the referees for their positive comments and have addressed them below. In addition we have been made some minor changes throughout the paper and added a new reference to a new HyIce 2018 study in the introduction (Vogel et al., 2024).

**1.  Reviewer Comment # 3**

RC: *The authors have written detailed answers to the reports of referee #1 and #2, which I consider as being well argued and sufficient. However, it is really important to highlight the work of Eufemio et al. 2023, also when it was not measured at the same place, but the results are very similar. So I would expect that the authors show respect and make clear that their report is second. Nevertheless, the measurement report is valuable and must be published, since it adds interesting information and knowledge to the discussion for the scientific community focusing on heterogeneous ice nucleation triggered by biological materials from boreal forests. From my point of view, the boreal forests are key for the understanding of heterogeneous ice nucleation in the troposphere.*

AR: *Thank you for your feedback. In addition to our frequent mention of the Eufemio et al. (2023) paper we have added new statements throughout the introduction to highlight that our work has come second. However, we do stress that our study is unique, with different objectives and a different methodology.*

> Adding to evidence for lichenous INPs, this study shows that lichen from a European Boreal forest in Hyytiälä harbour INPs. This novel finding may be especially important in this snow covered habitat where few, if any, other biological INP sources are available.

> In a recent study, Eufemio et al. (2023) tested lichens collected across Alaska for their ice nucleating ability, pointing to their possible impact on cloud glaciation in a warming Arctic.

> In this paper we tackle the first part of the hypothesis, namely the question of if the lichens in the forest during the winter of HyICE-2018 contained ice-nucleating entities. Eufemio et al. (2023) have made this possibility obvious since they showed that lichen from across Alaska harbour INPs. It remains to us to confirm this for the Hyytiälä boreal forest. A positive outcome would provide motivation to address the question of whether sufficient quantities of INPs are released into the atmosphere to influence the INP population and subsequent cloud formation.

**2.  Reviewer Comment # 4**

RC:  *Proske et al. have revised a manuscript based on INP measurements from several lichen species collected in Finland. The study presents results including size-resolved analysis via filtration and thermal treatments to assess the INP composition of the lichen samples. Although significant effort has been made in the revision process, several major issues remain that need to be addressed before the manuscript can be considered for publication.*

AR:  *Thank you for your substantial feedback, which we have incorporated as detailed in the following.*

**2.1.  General comments**

RC:  *Abstract: I understand the concern of the previous reviewers regarding the novelty of the work. The interesting aspect is that these lichens are present and consist of warm temperature INPs during winter when other more prominent/abundant biological sources are covered in snow, thus, could be important for influencing the wintertime airborne INP population. This concept should be highlighted more clearly, especially in the abstract, but additionally throughout.*

AR:  *We have rephrased the sentence in the abstract that contained this thought, to make it more explicit:*

> This novel finding may be especially important in this snow covered habitat where few, if any, other biological INP sources are available.

In addition to the introduction, where we state:

> However, the source of these INPs was unclear since the surface was snow covered, which rules out leaf litter or bare soil as sources of INPs. We hypothesize that the trees which bore lichen, that were exposed even in the winter when the canopy and the ground were snow covered (see Fig. 1), might have been a source of INPs.

we have added a reiteration of this point to the beginning of the Results section:

> Our hypothesis is that these biological INPs originate from the lichen that is abundant in the boreal forest ecosystem even when there is snow cover.

RC:  *Hypothesis: Shouldn't this hypothesis be more specific, based on what was actually tested? The study only tests winter samples. I would guess there is some level of dormancy in the winter and the samples might behave differently in the summer. Also, the study does not actually test the hypothesis in that the presence of lichen is not investigated in the airborne INPs from HyICE-2018. Rather, the study tests the potential of local sources to contain INPs that could become airborne during the winter. The hypothesis should be rewritten to reflect what is actually verified/denied in this study and should be revisited in the discussion section.*

AR:  *In our effort to explain a) the hypothesis that lichen could provide a local source of INP in the snow-covered winter and b) our approach of testing the first part of it, namely whether local lichen contain INP, we have*

*apparently lost the reader. We have reformulated the corresponding section to make the distinction between the overall hypothesis and what we are able to contribute to addressing this hypothesis in this paper clearer:*

> We hypothesize that the trees which bore lichen, that were exposed even in the winter when the canopy and the ground were snow covered (see Fig. 1), might have been a source of INPs. In this paper we tackle the first part of the hypothesis, namely the question of if the lichens in the forest during the winter of HyICE-2018 contained ice-nucleating entities. Eufemio et al. (2023) have made this possibility obvious since they showed that lichen from across Alaska harbour INPs. It remains to us to confirm this for the Hyytiälän boreal forest. A positive outcome would provide motivation to address the question of whether sufficient quantities of INPs are released into the atmosphere to influence the INP population and subsequent cloud formation.

**RC:** *"Steps": It is very difficult to visually discern the steps in the current figures. Also, if trying to discuss different INP populations, it is best practice to use differential spectra. A couple of suggestions here: 1) make the current spectra bigger/wider so the small differences are easier to see and 2) make a figure with differential spectra, so that the two populations at -16C and -18C are evident.*

AR: *Thank you for the great suggestion! We have followed it by adding Fig. C1 that shows the differential spectra and included references to it in the text. Unfortunately not all samples had dilutions that lined up well enough to allow for a sensible interpolation, so we have only translated some of our data into differential spectra. However, the different INP species (peaks) corresponding to our 'steps' are clearly distinguishable.*

**RC:** *Filtrations: Why were the same filtrations not executed on all the samples? Specifically, sample B was not subject to the 0.02 μm as sample C was. For the individual species, only 2 μm filtration was done. It is difficult to intercompare the samples when the same filtrations were not done. The authors could either omit the 0.02 μm results or very clearly describe why it was only done on one sample. Also, why were the same filtrations not done on the individual species? That may have helped determine which are the most dominant INPs in the sample mixtures, by being able to compare similarly-manipulated samples.*

AR: *We agree that it would have been best and regret to not have results for all filter sizes and samples, but have added a description as you suggested:*

> Only for sample C all filter sizes were used, as it was realized after the processing of sample B that further size differentiation would be desirable. For the species specific tests the samples were partly too small to use all filters so only the 2 μm filter was used.

**RC:** *Proposed emission mechanisms: First, suspending lichen in a bunch of water would not replicate possible aerosolization methods in the real environment (lines 132-135). Suspension in water and atmospheric emission mechanisms could be completely different. This sentence should be omitted; the authors should instead state that there could be differences in the suspension solution versus what might actually become airborne. Second, the authors discuss how the smaller INPs could become airborne by adhering to large particles. Has this process been evaluated for lichen? What larger particles are present in lichen that these smaller INPs would stick to, that are subject to detaching from the lichen surface? I can see how this emission process is possible for soil surfaces, but it is not clear for lichen. Additional justification for these conclusions should be provided as the link between the results and possible emission processes is currently not clearly drawn.*

AR: *Regarding your first point, the lines you quote make the point that our sampling procedure improves upon previous studies because we are not washing the lichen prior to probing nor grinding it up. It continues to state that also our practice is clearly different from wind dispersal, which we have amended to be more clear:*

> *While this approach is clearly different from bioaerosol production from lichen via wind, it does bias the analysis towards the entities associated with lichen that are likely to become aerosolised.*

*Your second point is covered in the Discussion section in l.334 to 340 (in the previous manuscript version). We could imagine the smaller particles to be fragments of dispersal particles. They could attach to whole spores, soredia or isidia, which have been shown to become aerosolised. The particular mechanism of smaller INP entities sticking to these larger entities has not been studied for lichen as far as we know (see l. 364ff, which have now been moved to the introduction).*

RC: **Reordering text: The results section was hard to follow - it did not flow naturally. I suggest reordering, so that the current section 3.1 is first (describing the individual species) followed by the start of the results up to line 278 (more complex investigation of potentially realistic ambient external mixtures). The text on lines 278-285 and the corresponding figure (comparing the extraction techniques) is more of a methodology testing. I suggest moving this to the methods as it does not test the hypothesis or apply to any sort of emission process that would realistically happen.**

AR: *Thank you for the suggestion, which we were happy to follow. Note that we have moved Fig. 6 and its discussion to the Appendix instead of to the Methods section where we felt it could overwhelm the reader with the results contained in the figure not discussed yet.*

RC: **Along these lines, I found it unclear that B and C are called "samples" when in reality, they are roughly the same mixture just tested under different suspension techniques. (Side question: Why was one master sample not mixed and then aliquoted for the different extraction methods?) Perhaps just call them the same sample mixture, but in the methods, describe that they were tested differently. You could even call the spectra "hand shaken" and "rotary mixed" or something along those lines.**

AR: *We understand that this is a little difficult to follow, but the alternatives wouldn't necessarily be better. We want to do justice to the fact that these were two different 'samples', with different lichen samples (albeit collected at the same time and stored in the same bag) and extraction techniques. The fact that we have two samples is due to our finding that one could not simply draw more sample from the same suspension that still contained the lichen, as discussed in l. 247ff. Thus we needed to set up a fresh suspension where for sample C we took care to draw enough solution from the suspension for all filter tests we desired to do.*

RC: **In general, there are many qualitative comparisons drawn between the results presented here and previous studies, and for inter-sample comparisons (i.e., the use of "steps" or indicating spectral features). It would be a stronger paper if actual concentration values were compared and differential spectra were used for the study sample intercomparisons. And for the filtrations, it would be helpful to present the % of the total INPs that were $10\,\mu\text{m}$, $2\,\mu\text{m}$, $0.1\,\mu\text{m}$, and $0.02\,\mu\text{m}$.**

AR: *We included differential spectra in C1 (discussed above). We attempted to produce percentage remaining plots (see below), but we feel these plots do not add anything that the fraction frozen plots already show. They also introduce issues such as fractions above unity, that are related to poor counting statistics in some cases.*

AR: *Also, we have created a new plot to compare with literature data (see response to specific comment below).*

[Figure]

[Figure]

Figure 1: Relative contributions of the different size fractions to $n_m$ for sample B in **a)**, and sample C in **b)**, i.e. $\frac{n_{\mathrm{m,10\,\mu m}}}{n_{\mathrm{m,unfiltered}}}$. To make the data comparable and allow for a division, the spline fit from de Almeida Ribeiro et al. (2023) was employed and the results were interpolated linearly. For the $10\,\mu m$ size fraction of sample B, the point at $-14\,°C$ and $8.4 \times 10^4\,g^{-1}$ needed to be excluded to allow for a smooth fit.

**2.2. Discussion section**

**RC:** *A few things to point out here.*

**RC:** *1. Some of this section belongs in the intro (lines 343-356), as it does not actually help describe or explain the study results.*

AR: *We have moved the section to the introduction as suggested.*

**RC:** *2. This section would be much more impactful if there was a comparison figure shown, since there are a handful of previous studies evaluating INPs in lichen (something like spectral "ranges" as in Fig 1-10 of Kanji et al. (2017) or a bar plot of spectral ranges as in Fig 4 in Creamean et al. (2021)). It is difficult to put the current results into context without some sort of summarizing visual.*

AR: *We welcome this suggestion and have added a new figure to the discussion section. We took the time to digitise the old data from the Kieft papers and contacted Eufermio for their data rather than plotting ranges. The caveat with this comparison is that the preparation methods are different (which is why we refrained from doing it in the past), but we have included this caveat and discussed the figure appropriately. The discussion has been revised accordingly, but not reproduced here to be keep the response succinct. The fact that our activity is at the low end of the reported activities may well be related to the sample preparation method.*

**RC:** *3. Intercomparing onset freezing temperatures with previous studies using different instruments is not recommended. Instrumental limitations and detection limits can yield different ranges of freezing onsets. Try to refrain from onset comparisons unless doing so with just the samples in the current study.*

AR: *We agree and have followed the referee's advice and removed that comparison. Onset comparisons can be logical if the activity is very steeply temperature dependent, but this is not a given here, so justification of the comparison would need to be given (which would distract from the main points we are making).*

**RC:** *4. The main idea behind this study is to test what local sources might have been observed in the air. Thus, putting these findings into the context of those from the air during HyICE-2018 should be included here. This text basically exists on lines 369-377 in the conclusion section, but I suggest moving it to the discussion.*

AR: *We have followed your suggestion and moved that paragraph to the discussion section.*

RC: **5. Along these lines, what other sources could be possible during the winter? What about anything on the needle surfaces or resuspension from the snowpack? I realize these were not tested here, but other realistic possibilities should be mentioned and cited.**

AR: *Yes, these other potential processes were not tested here so we do not want to discuss them in detail. We have inserted a brief mention of them in the discussion section.*

> Alternatively, the biological INP observed during HyIce-2018 might have come from a different source. Possibilities include release of INP from the needles or other surfaces of pine trees (Seifried et al., 2023) or perhaps from blowing snow that might release aerosol if snow particles sublime (Frey et al., 2020).

**2.3. Specific comments**

RC: *Line 3: Specify that these were ambient/airborne INPs, not to be confused with the current work.*

AR: *We have specified this in the abstract as follows:*

> During the HyICE-2018 campaign, which took place in the boreal forest of Hyytiälä, substantial concentrations of airborne, heat sensitive biological INPs were observed despite many potential biological sources of INPs being snow covered. A potential source of INPs that were not covered in snow were lichens that grow on trees, hence we investigated these lichens as a potential source of biological INPs in this boreal forest environment.

RC: **Lines 7-8: The abstract does not entirely reflect the actual findings. The authors report on the 0.02 μm results being the most substantial, but filtration was only done on one of the samples. In the paper, several times (e.g., lines 256-257, the discussion, and the conclusions), it is indicated that most of the INPs were in the 0.1-10 μm range. That should be reported here instead.**

AR: *We remain with our point that activity remains at small sizes, but have agreed to highlight the $0.1\,\mu\mathrm{m}$ filter size instead as it is tested more in our study.*

RC: **Lines 44-46: Why is one study reported in onset freezing range and one in median? It is best to compare apples to apples and report median or another common value for both. (Intercomparing onset temperatures between different techniques is not advised; see comment above.)**

AR: *That's because the two studies only give us onsets and medians and no common value, which we agree is regrettable.*

RC: **Lines 120-121: Why stored at room temperature and not at least close to the temperature in which the samples were collected? The authors should provide a statement about the caveats in possible changes to the samples while stored at room temperature. Often, vegetation samples are stored in a refrigerator (something like 4C) or even frozen. The authors cite Stopelli et al. (2014) which describe that certain conditions are known to activate the ice-nucleating activity of bacterial cells. While they are referring to samples in snow water, bacterial activity can change when the temperature of the air increases (i.e., what can happen in the summer compared to winter samples).**

AR: *We have added a sentence to make this caveat explicit. However, we also note that storing in a sealed bag at*

*room temperature meant the samples were stored at low relative humidity, which tends to inhibit biological activity. Nevertheless, storage is a problem that we generally face and is unsatisfactorily resolved. Freezing is known to degrade samples, as is storing in a refrigerator, so these standard approaches are not necessarily suitable.*

> By storing at room temperature, the samples were preserved at low relative humidity, conditions under which biological activity is inhibited. Nevertheless, we note that the storage at room temperature was pragmatic, and it is possible that the activity of the samples might be somewhat dependent upon the storage conditions.

**RC:**   ***Table 1: Why were these particular percentages of species chosen for B and C? There needs to be some justification provided in the text.***

**AR:**   *We have added a sentence clarifying that the percentages were chosen to mimic the perceived concentrations in the bag.*

**RC:**   ***Line 320: Why is a quantifiable comparison not possible?***

**AR:**   *As you state above, that is because onset freezing temperatures cannot be compared between setups. We have followed your advice above and removed that section now.*

**RC:**   ***Line 322: Saying they "also" found high variability does not align with the current study. The species presented in Fig 7a look pretty similar. But this is also admittedly subjective. Making this statement more quantifiable, for example by using X orders of magnitude spread at select temperatures, and comparing the same to Eufemio et al. would get the point across, if indeed there was high variability found in both that study and the current one. If not found, why might this study not have "high variability"?***

**AR:**   *We have made that statement more precise.*

> They also found high variability in ice nucleating activity between species of lichen ($T_{50}$ of -8 and $-15\,°\mathrm{C}$ for the boreal samples; compare to Fig. B1) as well as sensitivity to heat.

**RC:**   ***Lines 335-336: What are the typical sizes of whole spore soredia or isidia? Some spores can have sizes down to $2\,\mu\mathrm{m}$ but I am not certain about these two specifically.***

**AR:**   *According to Bowler and Rundel (1975), soredia are $25$ to $100\,\mu\mathrm{m}$ in diameter; and isidia are between $10$ and $300\,\mu\mathrm{m}$ in diameter and $500$ and $3000\,\mu\mathrm{m}$ in height.*

**References**

Bowler, P. A. and P. W. Rundel (June 1975). "Reproductive Strategies in Lichens". In: *Botanical Journal of the Linnean Society* 70.4, pp. 325–340. ISSN: 00244074. DOI: `10.1111/j.1095-8339.1975.tb01653.x`.

Creamean, Jessie M., Julio E. Ceniceros, Lilyanna Newman, Allyson D. Pace, Thomas C. J. Hill, Paul J. DeMott, and Matthew E. Rhodes (June 2021). "Evaluating the Potential for Haloarchaea to Serve as Ice Nucleating Particles". In: *Biogeosciences* 18.12, pp. 3751–3762. ISSN: 1726-4170. DOI: `10.5194/bg-18-3751-2021`.

de Almeida Ribeiro, Ingrid, Konrad Meister, and Valeria Molinero (May 2023). "HUB: A Method to Model and Extract the Distribution of Ice Nucleation Temperatures from Drop-Freezing Experiments". In: *Atmospheric Chemistry and Physics* 23.10, pp. 5623–5639. ISSN: 1680-7316. DOI: 10.5194/acp-23-5623-2023.

Eufemio, Rosemary J, Ingrid de Almeida Ribeiro, Todd L Sformo, Gary A Laursen, Valeria Molinero, Janine Fröhlich-Nowoisky, Mischa Bonn, and Konrad Meister (2023). "Lichen Species across Alaska Produce Highly Active and Stable Ice Nucleators". In: *Biogeosciences* 20. DOI: 10.5194/bg-20-2805-2023.

Frey, M. M. et al. (2020). "First direct observation of sea salt aerosol production from blowing snow above sea ice". In: *Atmospheric Chemistry and Physics* 20.4, pp. 2549–2578. DOI: 10.5194/acp-20-2549-2020. URL: https://acp.copernicus.org/articles/20/2549/2020/.

Kanji, Zamin A., Luis A. Ladino, Heike Wex, Yvonne Boose, Monika Burkert-Kohn, Daniel J. Cziczo, and Martina Krämer (Jan. 2017). "Overview of Ice Nucleating Particles". In: *Meteorological Monographs* 58, pp. 1.1–1.33. ISSN: 0065-9401. DOI: 10.1175/AMSMONOGRAPHS-D-16-0006.1.

Seifried, Teresa M., Florian Reyzek, Paul Bieber, and Hinrich Grothe (2023). "Scots Pines (Pinus sylvestris) as Sources of Biological Ice-Nucleating Macromolecules (INMs)". In: *Atmosphere* 14.2. ISSN: 2073-4433. DOI: 10.3390/atmos14020266. URL: https://www.mdpi.com/2073-4433/14/2/266.

Vogel, F. et al. (2024). "Ice-nucleating particles active below −24 °C in a Finnish boreal forest and their relationship to bioaerosols". In: *EGUsphere* 2024, pp. 1–25. DOI: 10.5194/egusphere-2024-853. URL: https://egusphere.copernicus.org/preprints/2024/egusphere-2024-853/.

This document was generated with a layout template provided by Martin Schrön (github.com/mschroen/review_response_letter).

---

## Author Response (AR3)

**Author Response to Reviews of**

**Measurement report: The ice-nucleating activity of lichen sampled in a northern European boreal forest**

Ulrike Proske, Michael P. Adams, Grace C. E. Porter, Mark Holden, Jaana Bäck, and Benjamin J. Murray
*ACP,* `doi:10.5194/egusphere-2023-2780`

RC: *Reviewer Comment*,     AR: *Author Response*,     ☐ Manuscript text

**1. Reviewer Comment # 5, from Reviewer # 4**

RC: *Proske et al. have significantly revised the manuscript, improving its clarity and strengthening the conclusions. The manuscript now presents a cohesive narrative, and the new additions have made the findings more robust. This is highly engaging work, and I believe the manuscript is now suitable for publication. I have only a few minor suggestions for the authors to consider before final acceptance.*

AR: *We thank the referee for reviewing the manuscript a second time and again offering constructive feedback.*

RC: *The novelty of this study lies not only in its location but also in its detailed investigation of INP size. While Kieft and Ruscetti (1990) examined only $0.2\,\mu\mathrm{m}$ INPs, this study expands on that by presenting data from filtration across multiple size ranges. Understanding the size distribution of INPs is undoubtedly valuable information. The authors might consider emphasizing this aspect more explicitly in both the abstract and the conclusion of the introduction.*

AR: *We agree that the size filtration is a valuable addition of this study. While we believe the abstract highlights this aspect sufficiently ("Successive filtration to smaller sizes removes some of the most active INPs in suspension, but substantial activity remains even when filtering to $0.1\,\mu\mathrm{m}$."), we have added the point to the conclusion of the introduction as you suggested:*

> In this paper we tackle the first part of the hypothesis, namely the question of if the lichens in the forest during the winter of HyICE-2018 contained ice-nucleating entities. Eufemio et al. (2023) have made this possibility obvious since they showed that lichen from across Alaska harbour INPs. It remains to us to confirm this for the Hyytiälä boreal forest, and to investigate the size of the lichenous INPs that we find.

RC: *Line 11: Typo: "Hyytiäläre" should be corrected.*
*Line 12: Consider adding "previous" or "existing" before "evidence" for clarity.*

AR: *Thank you for pointing us to these oversights, which we have corrected accordingly.*

RC: *Line 114: I assume you mean above $-24\,°\mathrm{C}$ here?*

AR: *No, in fact Vogel et al. (2024) specifically report INPs below $-24\,°\mathrm{C}$.*

RC: *Lines 444-445: The statement, "Many of the lichenous INPs were found to be between 0.1 and $2\,\mu\mathrm{m}$ in size when immersed in water, and those active at temperatures higher than $18\,°\mathrm{C}$ were heat-labile," raises an*

*interesting point. Could some of these particles be bacteria living symbiotically with the lichen? Bacterial cells typically range around $\approx 1\,\mu\mathrm{m}$ and are often heat-labile. While the authors mention that Kieft (1988) did not find cultivable bacteria in lichen samples, it's important to note that not all ice-nucleating bacteria are easily cultured. The introduction touches on how bacteria, among other things, can contribute to the INPs from lichen samples. Given the size range and heat sensitivity, the authors might want to discuss the potential bacterial connection to their findings, specifically, further in their discussion.*

AR: *Thank you for this suggestion. We have taken up your discussion idea in the Conclusions section by adding:*

> Alternatively, the INP may also be bacteria living symbiotically with the lichen. As mentioned in Sec. 1, the bacteria that Kieft (1988) cultivated from lichen showed no ice-nucleating activity, but not all ice-nucleating bacteria are easily cultured.

**References**

Eufemio, Rosemary J, Ingrid de Almeida Ribeiro, Todd L Sformo, Gary A Laursen, Valeria Molinero, Janine Fröhlich-Nowoisky, Mischa Bonn, and Konrad Meister (2023). "Lichen Species across Alaska Produce Highly Active and Stable Ice Nucleators". In: *Biogeosciences* 20. DOI: 10.5194/bg-20-2805-2023.

Kieft, Thomas L (1988). "Ice Nucleation Activity in Lichens". In: *Applied and Environmental Microbiology* 54, p. 5.

Vogel, F. et al. (2024). "Ice-nucleating particles active below $-24\,°\mathrm{C}$ in a Finnish boreal forest and their relationship to bioaerosols". In: *EGUsphere* 2024, pp. 1–25. DOI: 10.5194/egusphere-2024-853. URL: https://egusphere.copernicus.org/preprints/2024/egusphere-2024-853/.

This document was generated with a layout template provided by Martin Schrön (github.com/mschroen/review_response_letter).